# CAPro: Webly Supervised Learning with Cross-Modality Aligned Prototypes

**Yulei Qin**[1]    **Xingyu Chen**[2]    **Yunhang Shen**[1]    **Chaoyou Fu**[1]

**Yun Gu**[3]    **Ke Li**[1]    **Xing Sun**[1]    **Rongrong Ji**[4]

[1]Tencent YouTu Lab    [2]ByteDance
[3]Shanghai Jiao Tong University    [4]Xiamen University
`yuleiqin@tencent.com`

## Abstract

Webly supervised learning has attracted increasing attention for its effectiveness in exploring publicly accessible data at scale without manual annotation. However, most existing methods of learning with web datasets are faced with challenges from label noise, and they have limited assumptions on clean samples under various noise. For instance, web images retrieved with queries of "*tiger cat*" (a cat species) and "*drumstick*" (a musical instrument) are almost dominated by images of tigers and chickens, which exacerbates the challenge of fine-grained visual concept learning. In this case, exploiting both web images and their associated texts is a requisite solution to combat real-world noise. In this paper, we propose Cross-modality Aligned Prototypes (CAPro), a unified prototypical contrastive learning framework to learn visual representations with correct semantics. For one thing, we leverage textual prototypes, which stem from the distinct concept definition of classes, to select clean images by text matching and thus disambiguate the formation of visual prototypes. For another, to handle missing and mismatched noisy texts, we resort to the visual feature space to complete and enhance individual texts and thereafter improve text matching. Such semantically aligned visual prototypes are further polished up with high-quality samples, and engaged in both cluster regularization and noise removal. Besides, we propose collective bootstrapping to encourage smoother and wiser label reference from appearance-similar instances in a manner of dictionary look-up. Extensive experiments on WebVision1k and NUS-WIDE (Web) demonstrate that CAPro well handles realistic noise under both single-label and multi-label scenarios. CAPro achieves new state-of-the-art performance and exhibits robustness to open-set recognition. Codes are available at `https://github.com/yuleiqin/capro`.

## 1 Introduction

Large-scale annotated datasets (*e.g.*, ImageNet [1]) were the driving force behind the revolution of computer vision in the past decade, but the expensive and time-consuming collection process now becomes the bottleneck of model scaling. Consequently, researchers seek to crawl web images, where search queries and user tags are directly used as labels. However, the large proportion of noise in web datasets (*e.g.*, 20% in JMT-300M [2], 34% in WebVision1k [3], and 32% in WebFG496 [4]) impedes learning visual concepts. Many studies on Webly Supervised Learning (WSL) are conducted to reduce the negative impact of noise and effectively explore web data [5, 6, 7, 8, 9, 10].

37th Conference on Neural Information Processing Systems (NeurIPS 2023).

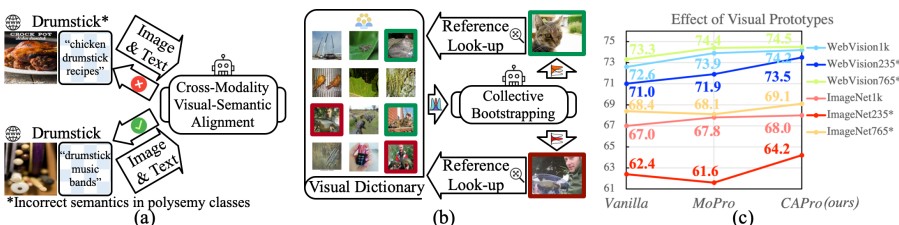

Figure 1: (a) We explore cross-modality alignment to select clean examples and generate visual prototypes with correct semantics. (b) Collective bootstrapping provides consistent label references and regularization from visual dictionary. (c) Compared to 765 unambiguous classes, our advantage is much more highlighted on 235 classes where semantic noise prevails due to polysemy concepts.

Early WSL methods claim that simply scaling up datasets with standard supervised learning suffices to overcome web noise [3, 11, 2, 12]. Such a solution comes at the cost of huge computation resources, and the supervision source from noisy labels is proved suboptimal [13, 10, 14]. Therefore, various techniques are designed to reduce noise [15], such as neighbor density [16], guidance of clean samples [17, 18, 19], confidence bootstrapping [20, 21, 22], and side information [23, 24].

Despite the promising improvement, the above-mentioned methods still face challenges. First, most of them address certain types of noise such as label-flipping noise and out-of-distribution (OOD), neglecting the critical-yet-under-explored **semantic noise**. To clarify, semantic noise is caused by the misalignment between image contents and the associated texts when the search query (*e.g.*, class) has multiple and ambiguous interpretations and the retrieved images do not correspond to the correct semantics. Without proper contextual information, it is rather difficult to pinpoint clean examples in polysemy classes. Second, the dominant idea of bootstrapping labels and discarding incorrect data is prone to noise overfitting [13]. The model predictions on individual images vary sharply over training epochs and such inconsistency also makes WSL inefficient. Some methods [4, 25, 26] also maintain peer models and require alternative steps to improve the reliability of bootstrapping. However, their complicated training procedures restrict scalability in practice.

To this end, we propose CAPro: **C**ross-modality **A**ligned **Pro**totypes for robust representation learning from web data. Compared with previous prototypical methods [19, 27, 28], CAPro is able to handle label-flipping noise, OOD, and especially the semantic noise that remains unexplored (see Fig. 1).

First, CAPro exploits web data across modalities to formulate *semantically-correct textual and visual prototypes*. Since visual prototypes simply formed with images suffer from semantic ambiguity, we propose **text matching** to leverage textual prototypes to establish their noise-robust estimation. Motivated by successful language models [29, 30, 31], we extract textual knowledge to imply the extent to which one instance is aligned with its textual prototype. Specifically, we prepare descriptive texts (*e.g.*, definitions) of each category and project them into the embedding space as textual prototypes via the pre-trained language model. For each sample, its affiliated texts of the website title, caption, and tags are also encoded into the same embedding space. The similarity between a sample and its prototype indicates "cleanness". Since incomplete and mismatched image-text pairs introduce additional noise [32], we bring in **text enhancement** with text guidance from visually-similar neighbors to mitigate the effect of noisy texts on clean sample selection. We consecutively construct image and text graphs and rerank neighbor candidates for text matching and enhancement. Samples that exactly match the target semantics are chosen as anchors for the initialization of visual prototypes. These visual prototypes are continuously polished up by high-quality web images to improve generalizability and discriminability. During representation learning, the intra-class distance between prototypes and instances is minimized by contrastive learning for regularization. By means of *class-representative visual prototypes*, various kinds of noise can be filtered out.

Second, we propose **collective bootstrapping** (CB) to provide smoother label reference by *extending bootstrapping with collective knowledge*. For each sample, instead of bootstrapping its target independently [33, 34], CAPro keeps bootstrapping the entire dynamic dictionary and provides label reference in the mode of dictionary look-up. The dictionary keys are the learned representations from the sampled data while the query is the current encoded sample. We aggregate model predictions on all keys and use their weighted combination as the pseudo target, where weights are determined by the matching scores of query-key pairs. By penalizing deviation from such targets, The proposed

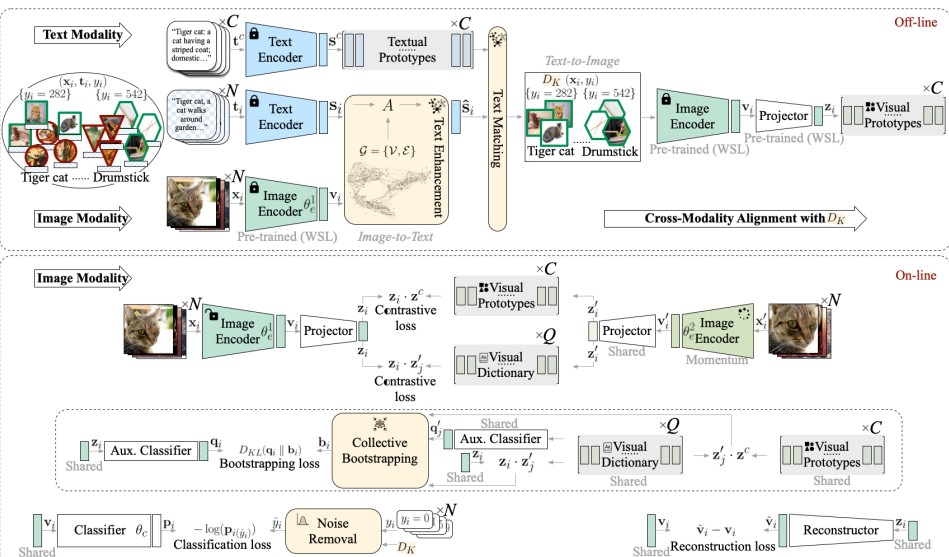

Figure 2: Overview of CAPro. Images $\mathbf{x}_i$ and texts $\mathbf{t}_i$ are respectively fed into the image and text encoders for features $\mathbf{v}_i$ and $\mathbf{s}_i$. Then, $\mathbf{v}_i$ is projected into the embedding space as $\mathbf{z}_i$, followed by the reconstruction from $\mathbf{z}_i$ to $\tilde{\mathbf{v}}_i$. Visual prototypes $\mathbf{z}^c$ are initialized with anchor instances that are selected by matching enhanced texts $\tilde{\mathbf{s}}_i$ to textual prototypes $\mathbf{s}^c$ for semantic alignment. They are constantly polished up by clean images and engage in contrastive learning to constrain cluster distribution. Collective bootstrapping exploits visual dictionary for regularization on the auxiliary classifier output $\mathbf{q}_i$, where each key embedding is matched to the query for the reference $\mathbf{b}_i$. Web labels $y_i$ are simultaneously refined as $\tilde{y}_i$ for "denoised" supervision on the classifier output $\mathbf{p}_i$.

CB achieves two advantages: 1) It encourages consistent performance among the query and visually similar keys. 2) Unstructured noise is suppressed by referring to the dictionary for label regularization. Our CB can also be viewed as encoding the neighborhood structure of data in the low-dimensional space, where the closely matched keys are neighbors of the query. Inductive propagation of self-labels is implicitly realized through such a structure, which draws on the assumption of manifold regularization [35] that close samples probably belong to the same class.

In summary, our contributions are summarized as:

- We propose CAPro, a prototypical learning framework to efficiently handle various noises including label-flipping noise, OOD, and semantic noise for webly supervised learning.

- We integrate class prototypes across text and image modalities to align with unambiguous semantics. We verify that text enhancement by visual guidance is the key to handling noisy texts for clean sample selection, which in turn improves visual prototypes.

- We investigate collective bootstrapping as label regularization by matching one query to all keys in a dynamic dictionary for reference. We show that scaling up the nearest neighbors from the mini-batch to the entire dictionary better leverages the visual data structure.

Experiments on WebVision1k and NUS-WIDE (Web) confirm the competitiveness of CAPro with prior state-of-the-art methods. CAPro performs robustly against real-world noise under single-label and multi-label scenarios and demonstrates superiority in open-set recognition.

## 2   Related Work

**Webly Supervised Learning (WSL)**   WSL aims at utilizing abundant but weakly labeled web data. It serves a range of tasks, such as recognition [36, 37, 38, 39, 40, 41], detection [42, 43], and segmentation [44, 45]. In this study, we focus on visual representation learning [46, 11, 2, 3, 47, 12]. To combat noise, previous studies combine web labels with the pseudo labels generated by the model. With respect to the pseudo labels, Hinton *et al.* [34] adopts soft targets in a fashion of distillation.

Tanaka *et al*. [21] considers both soft and hard targets and proposes an alternative optimization framework. Yang *et al*. [22] estimates the correctness of hard targets on a case-by-case basis and dynamically balances the two supervision sources with label confidence. Recently, the idea of prototypical learning [48] has been applied for WSL. Han *et al*. [49] predicts self-labels by prototype voting and uses a constant ratio to combine web and pseudo labels. Momentum Prototype (MoPro) [28] improves prototypes with the momentum update policy [50] for smooth label adjustment.

The differences between CAPro and the closely related MoPro exactly highlight our contributions. First, we take advantage of both textual and visual prototypes to handle semantic misalignment noise. MoPro neglects such noise and its prototypes could be overwhelmed by irrelevant samples, which impairs the subsequent sample correction. Second, MoPro does not refer to appearance-similar samples for self-labels, which is prone to real-world noise that causes unreasonable label updates. In contrast, CAPro adopts bootstrapping wisely by performing dictionary look-up: the model prediction of the current query sample refers to its matching keys in the dictionary, where the matching score by visual similarity determines the contribution of each key. Third, MoPro only tackles single-label representation learning while CAPro extends prototypical contrastive learning for the multi-label scenario, which is non-trivial due to the intricate nature of noisy multi-labeled data. In that case, CAPro maintains prototypes in subspaces of the shared embedding space, which not only resolves the inter-class contrastive conflicts but also fosters implicit exploitation of label dependency.

**Noise-Robust Learning from Neighbors**   Several approaches attempt to correct labels with neighborhood consensus. Both Wang *et al*. [51] and Guo *et al*. [16] measure data complexity using local neighbor density for sample selection. Huang *et al*. [52] progressively discovers neighbors to form decision boundaries in an unsupervised manner. Bahri *et al*. [53] filters out training data whose label collides with the predictions of its $k$-NN. Wu *et al*. [54] employs topology to only keep the largest connected component of a $k$-NN graph. Neighborhood collective estimation [55] evaluates model confidence on the "cleanness" of each candidate by its neighbors. Ortego *et al*. [56] identifies correct examples by comparing original labels with the soft labels from their neighbors. Neighbors are also involved in consistency regularization [57, 58] to encourage similar outputs on samples within the $k$-neighborhood. Such inductive label propagation allows correct supervision to transfer directly to mislabeled data via the neighborhood structure.

Unfortunately, the aforementioned methods are not scalable due to the huge complexity of updating a global $k$-NN graph frequently. Both Li *et al*. [57] and Iscen *et al*. [58] only consider neighbors within a mini-batch for on-the-fly graph construction. However, web data tends to be sparsely distributed and the graph built within mini-batch samples hardly provides reliable neighbors. To deal with such a trade-off, CAPro pinpoints all potential neighbors in the dictionary by matching representations without explicit graph building. We maintain the dictionary as a queue whose length is much larger than the batch size, enabling collective bootstrapping from appropriate neighbors.

Nearest neighbors play a vital role throughout our CAPro, from text enhancement to matching and collective bootstrapping. Compared with previous methods, our mechanism differs in that: 1) We acquire guidance from cross-modality neighbors, where noisy texts are enhanced by image neighbors to alleviate the mismatch problem. In contrast, existing studies investigate neighbors of one modality. 2) We exploit reciprocal structures to filter nearest neighbors for pertinent text matching, while most works neglect those top-ranked false positive neighbors. 3) We resort to neighbors for on-line collective bootstrapping in a manner of dictionary look-up instead of explicit graph construction.

**Learning with Visual-Semantic Alignment**   Various tasks seek to learn visual representations for semantic concepts, including retrieval [59, 60, 61, 62, 63], caption [64, 65, 66], matching [67, 68], visual question answer [69], and zero-shot learning [70, 71]. Recently, learning unified embeddings has been studied for foundation models by language-image pre-training [72, 73, 74, 75, 76, 77].

In WSL, textual metadata such as titles and hashtags are too scarce to carry out language-image pre-training. Few studies harness both images and texts to learn semantically-correct representations. Zhou *et al*. [23] designs a co-training scheme to extract semantic embeddings to transfer knowledge from head to tail classes. Cheng *et al*. [24] builds visual and textual relation graphs to choose prototypes by graph-matching. Yang *et al*. [78] builds a visual-semantic graph and uses a graph neural network for label refinement. Nevertheless, these methods do not take noisy texts into serious consideration and underestimate their negative effect on seed selection. We also find that image outliers are wrongly kept even with textual concept matching. For example, images of a rugby team

are ranked top for the class "tiger-cat" just because the team name "tiger-cat" is frequently mentioned. On the contrary, CAPro introduces text enhancement by smoothing and reranking to improve its robustness to noise. Furthermore, the prototypes in [24, 78] are fixed during training, while CAPro keeps polishing up prototypes with clean examples for better domain generalizability.

**Noisy Correspondence Rectification**   One paradigm similar to WSL is noisy correspondence rectification or calibration [60, 65, 62, 66, 61, 63, 68, 79]. It tackles the mismatched image and text pairs and aims to simultaneously learn aligned visual and textual embeddings for improved cross-modal retrieval. Huang *et al.* [65] utilizes the memorization effect of neural networks to partition clean and noisy data and then learns to rectify correspondence. Hu *et al.* [61] derives a unified framework with contrastive learning to reform cross-modal retrieval as an N-way retrieval. Han *et al.* [66] proposes a meta-similarity correction network to view the binary classification of correct/noisy correspondence as the meta-process, which facilitates data purification. Our CAPro differs in two aspects: 1) We focus on the label noise where images are wrongly-labeled by keywords or hashtags. Noisy correspondence emphasizes the instance-level mismatch between an image and its associated text. 2) We aim to learn visual representations with categorical labels while most methods on noisy correspondence align image and text embeddings to improve cross-modal retrieval.

## 3   Method

### 3.1   Problem Definition and Framework Architecture

Given an interested class $y_i \in \{1, ..., C\}$, web data are collected as $D = \{(\mathbf{x}_i, \mathbf{t}_i, y_i)\}_{i=1}^N$, where $\mathbf{x}_i$ and $\mathbf{t}_i$ respectively denote the image and textual metadata. Due to the noise issues, $y_i$ might not equal to the ground-truth $y_i^*$. We aim to optimize a deep model $\mathcal{F}(\theta_e; \theta_c)$ with parameters of an encoder $\theta_e$ and a classifier $\theta_c$. Existing WSL studies often neglect $\mathbf{t}_i$ and seldom consider the intervention between images and texts. Comparatively, our CAPro unearths $\mathbf{t}_i$ for aligning visual representations with semantic concepts, which facilitates correction of various kinds of noise.

CAPro consists of the following components (see Fig. 2). **Siamese image encoders** extract features $\mathbf{v}_i, \mathbf{v}'_i \in \mathbb{R}^{d_v}$ from inputs $\mathbf{x}_i$ and their augmented counterparts $\mathbf{x}'_i$. Following MoCo [50], parameters of the first query encoder $\theta_e^1$ are updated by back-propagation and those of the second key encoder $\theta_e^2$ are updated by the momentum method. **A text encoder** generates embeddings $\mathbf{s}_i, \mathbf{s}^c \in \mathbb{R}^{d_t}$ respectively from the instance $\mathbf{t}_i$ and the category $\mathbf{t}^c$. Any off-the-shelf language model can be used with its pre-trained encoder frozen. **A classifier**, via a fully-connected (FC) layer, maps $\mathbf{v}_i$ to predictions $\mathbf{p}_i \in \mathbb{R}^C$ over $C$ classes. **A projector** distills discriminative low-dimensional embeddings $\mathbf{z}_i \in \mathbb{R}^{d_p}$ from $\mathbf{v}_i$. It has two FC layers, followed by $\ell_2$-normalization for unit-sphere constraint on $\mathbf{z}_i$. **A reconstructor**, symmetric to the projector, recovers $\tilde{\mathbf{v}}_i$ from $\mathbf{z}_i$ to be close to $\mathbf{v}_i$. **An auxiliary classifier**, of the same structure as the classifier, outputs predictions $\mathbf{q}_i \in \mathbb{R}^C$ on $\mathbf{z}_i$. **A dictionary**, implemented as a queue of size $Q \times d_p$, records keys for both contrastive learning and collective bootstrapping. The latest embeddings $\mathbf{z}'_i$ are enqueued while the oldest are dequeued.

Image encoders and classifiers are trained with a cross-entropy loss. Since features $\mathbf{v}_i$ contain redundant description that is vulnerable to image corruption and domain gap, we emphasize class-indicative contents by learning a low-dimensional embedding space. Inspired by denoising autoencoders [80, 27], a projector and a reconstructor are designed to optimize the projection from $\mathbf{v}_i$ to $\mathbf{z}_i$. An auxiliary classifier helps retain the representation capacity of $\mathbf{z}_i$.

$$\mathcal{L}_i^{\text{cls}} = -\log(\mathbf{p}_{i(y_i)}), \ \mathcal{L}_i^{\text{prj}} = \|\tilde{\mathbf{v}}_i - \mathbf{v}_i\|_2^2 - \log(\mathbf{q}_{i(y_i)}). \tag{1}$$

### 3.2   Cross-Modality Alignment

**Text Encoding**   For raw texts in metadata, we remove all html tags, file format extensions, punctuations, digits, and stop words. Then, tokenization is performed in accordance with the language model in use. After that, we obtain the pre-processed metadata $\mathbf{t}_i$ and use the text encoder to extract $\mathbf{s}_i$.

**Text Enhancement**   To handle missing and mismatched texts, we assume that similar images should share similar texts, and consider text enhancement with guidance from visual data structure. One simple way of encoding visual structure is to build a global $k$-NN graph on $\mathbf{v}_i$ [78]. However, our

preliminary experiments show that the top-ranked neighbors may not be pertinent due to the noise and domain gap. To detect *truly matched neighbors*, we construct a $k$-reciprocal-NN graph [81] $\mathcal{G} = \{\mathcal{V}, \mathcal{E}\}$ and use the re-ranking technique to evaluate neighbor relationship. Each node in $\mathcal{V}$ denotes an image and the edge connectivity from $\mathcal{E}$ is represented as the adjacency matrix $A$.

$$A_{ij} = \begin{cases} 1 - d(\mathbf{v}_i, \mathbf{v}_j) & , \text{if } \mathbf{x}_i \in \mathcal{R}(\mathbf{x}_j, k) \text{ or } \mathbf{x}_j \in \mathcal{R}(\mathbf{x}_i, k), \\ 0 & , \text{otherwise}, \end{cases} \tag{2}$$

where $\mathcal{N}(\mathbf{x}_i, k)$ and $\mathcal{R}(\mathbf{x}_i, k) = \{\mathbf{x}_j | \mathbf{x}_j \in \mathcal{N}(\mathbf{x}_i, k) \wedge \mathbf{x}_i \in \mathcal{N}(\mathbf{x}_j, k)\}$ respectively denote $k$-NN and $k$-reciprocal-NN of $\mathbf{x}_i$. The cosine distance is used here: $d(\mathbf{v}_i, \mathbf{v}_j) = 1 - \frac{\mathbf{v}_i \cdot \mathbf{v}_j}{\|\mathbf{v}_i\|\|\mathbf{v}_j\|}$. Neighbor re-ranking is achieved by re-calculating the pairwise distance. The vanilla cosine distance only weighs relative priority by measuring features, overlooking the context information of overlapped reciprocal neighbors. Hence, Jaccard Distance [81, 82, 83] is introduced to measure the intersection between reciprocal neighbors. The refined distance $d^*(\mathbf{v}_i, \mathbf{v}_j) = \frac{1}{2}(d(\mathbf{v}_i, \mathbf{v}_j) + d_J(\mathbf{v}_i, \mathbf{v}_j))$:

$$d_J(\mathbf{v}_i, \mathbf{v}_j) = 1 - \frac{\sum_{k=1}^{N} \min(V_{\mathbf{v}_i, \mathbf{v}_k}, V_{\mathbf{v}_j, \mathbf{v}_k})}{\sum_{k=1}^{N} \max(V_{\mathbf{v}_i, \mathbf{v}_k}, V_{\mathbf{v}_j, \mathbf{v}_k})}, \quad V_{\mathbf{v}_i, \mathbf{v}_j} = \begin{cases} \exp(-d(\mathbf{v}_i, \mathbf{v}_j)) & , \text{if } \mathbf{x}_j \in \mathcal{R}(\mathbf{x}_i, k) \\ 0 & , \text{otherwise}. \end{cases} \tag{3}$$

Given $\mathcal{G}$, smoothing is performed on $\mathbf{S} = (\mathbf{s}_1, \mathbf{s}_2, ..., \mathbf{s}_N) \in \mathbb{R}^{N \times d_t}$ via graph convolution [84]: $\hat{\mathbf{S}} = \tilde{D}^{-\frac{1}{2}} \tilde{A} \tilde{D}^{-\frac{1}{2}} \mathbf{S}, \tilde{A} = A + I_N, \tilde{D}_{ii} = \sum_j \tilde{A}_{ij}$, where $\tilde{A}$ refers to the adjacency matrix with self-connection and $\tilde{D}$ is the diagonal degree matrix.

**Textual Prototypes** To establish textual prototypes, we do not estimate $\mathbf{s}^c$ from instances in one class (*e.g.*, average) considering the insufficient, noisy nature of metadata. Instead, we refer to WordNet [85] for the vocabulary hierarchy [86, 78, 24]. For the $c$-th class, we extract its definition in WordNet and expand context with its siblings (synonyms), children (hyponyms), and parents (hypernyms). Then, we get $\mathbf{t}^c$ and encode it for $\mathbf{s}^c$. Such prototypes $\mathbf{s}^c$ have two advantages: 1) It enriches semantic representations of classes. For instance, the comprehensive text of the class "tiger cat" is *a cat having a striped coat; domestic_cat, house_cat, felis_domesticus, felis_catus: any domesticated member of the genus Felis.* It provides disambiguation to web instances of "tiger-cat" (*medium-sized wildcat in Central South America*) and "tiger, cat" (*large feline of forests in most of Asia having a tawny coat with black stripes; endangered*). 2) It reveals the underlying inter-class relationship by language encoding. The structural information of class hierarchy is hard to infer from metadata instances but can be directly indexed in WordNet.

**Text Matching** With textual prototypes as queries, web instances with correct semantics can be retrieved by matching queries to their embeddings as keys. To improve precision, the same distance measurement in Eq. (3) for $k$-reciprocal-NN encoding is adopted to rerank the matched candidates. We sort samples by distance in an ascending order, and select the top-$K$ as clean set $D_K$.

$$D_K = D_K^1 \cup D_K^2 \cup ... \cup D_K^C, D_K^c = \{(\mathbf{x}_i, \mathbf{t}_i, y_i) | (y_i = c) \wedge (d^*(\hat{\mathbf{s}}_i, \mathbf{s}^c) \leq \sigma_K^c)\}, \tag{4}$$

where $\sigma_K^c$ denotes the $K$-th smallest distance in the $c$-th class.

**Visual Prototypes** $D_K$ plays an anchoring role in shaping visual prototypes. We initialize the $c$-th prototype $\mathbf{z}^c$ by averaging instances in $D_K^c$: $\hat{\mathbf{z}}^c = \frac{1}{K} \sum_{\mathbf{x}_i \in D_K^c} \mathbf{z}_i, \mathbf{z}^c = \frac{\hat{\mathbf{z}}^c}{\|\hat{\mathbf{z}}^c\|_2}$. Given such a good starting point, visual prototypes are consistently polished up by trustworthy web examples with a momentum coefficient $m_p$: $\hat{\mathbf{z}}^c = m_p \mathbf{z}^c + (1 - m_p)\mathbf{z}_i, \mathbf{z}^c = \frac{\hat{\mathbf{z}}^c}{\|\hat{\mathbf{z}}^c\|_2}$. We perform instance-prototype contrastive learning to pull instances around their prototypes and push apart different class clusters. Instance-level discrimination is also encouraged to improve separation across classes.

$$\mathcal{L}_i^{\text{pro}} = -\log \frac{\exp(\mathbf{z}_i \cdot \mathbf{z}^{y_i} / \tau)}{\sum_{c=1}^{C} \exp(\mathbf{z}_i \cdot \mathbf{z}^c / \tau)}, \quad \mathcal{L}_i^{\text{ins}} = -\log \frac{\exp(\mathbf{z}_i \cdot \mathbf{z}_i' / \tau)}{\sum_{j=1}^{Q} \exp(\mathbf{z}_i \cdot \mathbf{z}_j' / \tau)}, \tag{5}$$

where $\tau$ is a temperature coefficient.

**Noise Removal** Noisy instances can be filtered out by self-prediction and instance-prototype similarity. We refer to MoPro [28] for rules of label adjustment by $\mathbf{o}_i \in \mathbb{R}^C$:

$$\mathbf{o}_i = \alpha \mathbf{p}_i + (1-\alpha)\mathbf{r}_i, \ \mathbf{r}_{i(k)} = \frac{\exp(\mathbf{z}_i \cdot \mathbf{z}^k/\tau)}{\sum_{c=1}^{C} \exp(\mathbf{z}_i \cdot \mathbf{z}^c/\tau)}, \tag{6}$$

where $\alpha$ balances two terms. Given a threshold $0 \leq \gamma \leq 1$, the pseudo-label $\hat{y}_i$ is estimated by:

$$\hat{y}_i = \begin{cases} y_i & \text{if } \mathbf{x}_i \in D_K, \\ \arg\max_c \mathbf{o}_{i(c)} & \text{else if } \max_c \mathbf{o}_{i(c)} > \gamma, \\ y_i & \text{else if } \mathbf{o}_{i(y_i)} > 1/C, \\ \text{Null } (OOD) & \text{otherwise.} \end{cases} \tag{7}$$

The above control flow guarantees continuous guidance from $D_K$ on cluster separation. If the highest score of $\mathbf{o}_i$ is above $\gamma$, the label will be changed accordingly. To prevent aggressive elimination of hard examples, we keep an instance till the next epoch so long as its confidence is above average. Otherwise, it is removed as OOD. The refined label $\hat{y}_i$ successively affects Eqs. (1) and (5).

### 3.3 Collective Bootstrapping

Due to memorization [87], as the training epoch increases, deep models will be prone to overfit noise even with the carefully designed logic of noise removal. We assume that overfitting occurs less dramatically when a majority can be consulted for the model to avoid over-confident decision on one single instance. With regard to the consultancy basis, the low-dimensional embedding is a good choice because its distilled description about visual contents is robust enough. Therefore, we propose to exploit the large dictionary, which is originally set up for instance-wise contrastive learning, to realize collective bootstrapping by dictionary look-up. The matching scores of the current query $\mathbf{z}_i$ to all keys $\mathbf{z}'_j$ in the dictionary act as the weights for the bootstrapped representations $\mathbf{b}_i \in \mathbb{R}^C$.

$$\mathbf{b}_i = \sum_{j=1}^{Q} w_{ij}(\alpha \mathbf{q}'_j + (1-\alpha)\mathbf{r}'_j), \ w_{ij} = \frac{\exp(\mathbf{z}_i \cdot \mathbf{z}'_j/\tau)}{\sum_{j=1}^{Q} \exp(\mathbf{z}_i \cdot \mathbf{z}'_j/\tau)}, \ \mathbf{r}'_{j(k)} = \frac{\exp(\mathbf{z}'_j \cdot \mathbf{z}^k/\tau)}{\sum_{c=1}^{C} \exp(\mathbf{z}'_j \cdot \mathbf{z}^c/\tau)}. \tag{8}$$

We minimize the difference between predictions and bootstrapping targets via a KL-divergence loss.

$$\mathcal{L}_i^{\text{bts}} = D_{KL}(\mathbf{q}_i \parallel \mathbf{b}_i) = \sum_{c=1}^{C} \mathbf{q}_{i(c)} \log \frac{\mathbf{q}_{i(c)}}{\mathbf{b}_{i(c)}}. \tag{9}$$

It not only allows collaborative contribution to individual soft label estimation, but also encourages consistent performance on visually similar examples. Note that such regularization is imposed on the auxiliary classifier $\mathbf{q}_i$. Compared with $\mathbf{p}_i$, constraints on $\mathbf{q}_i$ coincide with our contrastive learning setting without potential conflicts with the hard label assignment in Eq. (7). The total objective is:
$\mathcal{L} = \sum_{i=1}^{N}(1-\lambda^{\text{bts}})\mathcal{L}_i^{\text{cls}} + \lambda^{\text{bts}}\mathcal{L}_i^{\text{bts}} + \lambda^{\text{prj}}\mathcal{L}_i^{\text{prj}} + \lambda^{\text{pro}}\mathcal{L}_i^{\text{pro}} + \lambda^{\text{ins}}\mathcal{L}_i^{\text{ins}}.$

## 4 Experiments

### 4.1 Experimental Setup

We evaluate CAPro on WebVision1k [3] (Google500 [78]) and NUS-WIDE (Web) [88] for single-label and multi-label representation learning, respectively. They contain image-text pairs which are in line with our WSL setting. All datasets under investigation are described in Sec. A. We perform ablation studies on Google500 and NUS-WIDE for low cost without losing generalization [22, 78]. The R50 [89]/MiniLM (L6) [30] are used as image/text encoders by default. Exhaustive details about hyper-parameters, implementation, and training are elaborated in Secs. B C and Algo. 1.

### 4.2 Comparison with the SOTA

Table 1 reports the top1/top5 accuracy of WebVision1k and Google500. Results of the SOTA methods trained and evaluated on the same datasets are quoted here. Due to different choices of image encoders and training strategies, prior methods may not be directly comparable. For example, VSGraph adopts

Table 1: Results on WebVision1k and Google500. Best/2nd best are marked bold/underlined.

| Method | Back-bone | WebVision1k Top1 | Top5 | ImageNet1k Top1 | Top5 | Google500 Top1 | Top5 | ImageNet500 Top1 | Top5 |
|---|---|---|---|---|---|---|---|---|---|
| MentorNet [17] | IRV2 [90] | 72.6 | 88.9 | 64.2 | 84.8 | – | – | – | – |
| Curriculum [16] | IV2 [91] | 72.1 | 89.1 | 64.8 | 84.9 | – | – | – | – |
| Multimodal [92] | IV3 [93] | 73.2 | 89.7 | – | – | – | – | – | – |
| Vanilla [22] | R50D [94] | 75.0 | 89.2 | 67.2 | 84.0 | 75.4 | 88.6 | 68.8 | 84.6 |
| SCC [22] | R50D | 75.3 | 89.3 | 67.9 | 84.7 | **76.4** | 89.6 | 69.7 | 85.3 |
| Vanilla[†] [78] | R50 | 74.2 | 89.8 | 68.2 | 86.2 | 66.9 | 82.6 | 61.5 | 78.8 |
| CoTeach [20, 78] | R50 | – | – | – | – | 67.6 | 84.0 | 62.1 | 80.9 |
| VSGraph[†] [78] | R50 | **75.4** | 90.1 | **69.4** | 87.2 | 68.1 | 84.4 | 63.1 | 81.4 |
| Vanilla [28] | R50 | 72.4 | 89.0 | 65.7 | 85.1 | – | – | – | – |
| SOMNet [8] | R50 | 72.2 | 89.5 | 65.0 | 85.1 | – | – | – | – |
| Curriculum [16] | R50 | 70.7 | 88.6 | 62.7 | 83.4 | – | – | – | – |
| CleanNet [18] | R50 | 70.3 | 87.7 | 63.4 | 84.5 | – | – | – | – |
| SINet [24] | R50 | 73.8 | **90.6** | 66.8 | 85.9 | – | – | – | – |
| NCR [58] | R50 | 73.9 | – | – | – | – | – | – | – |
| NCR[†] [58] | R50 | 75.7 | – | – | – | – | – | – | – |
| MILe [26] | R50 | 75.2 | 90.3 | 67.1 | 85.6 | – | – | – | – |
| MoPro [28] | R50 | 73.9 | 90.0 | 67.8 | 87.0 | – | – | – | – |
| Vanilla (ours) | R50 | 72.6 | 89.7 | 67.0 | 86.8 | 69.9 | 86.5 | 64.5 | 83.1 |
| CAPro (ours) | R50 | 74.2 | 90.5 | 68.0 | **87.2** | 76.0 | **91.3** | **72.0** | **89.2** |

[†] Results on WebVision1k are under optimized training settings with batch size of 1024.

the same R50, but is trained with a batch-size of 1024. The benefits of a larger batch size have been validated in NCR, where the same method achieves 75.7% and 73.9% in top1 accuracy respectively for the batch size of 1024 and 256. We believe batch size is the reason that a vanilla baseline [78] surpasses most SOTAs. Due to the limited budget, training with a batch size of 1024 is currently not affordable, but we will experiment in the future. In this case, methods within each row group of Table 1 are fairly comparable with each other, including the vanilla trained by a cross-entropy loss.

CAPro achieves quite competitive performance on WebVision1k, with an improvement of 1.6% (top1 accuracy) over our vanilla. Although SCC and VSGraph respectively opt for stronger backbones (*e.g.*, R50D) and longer training epochs (*e.g.*, 150) with a larger batch size (*e.g.*, 1024), CAPro still excels in terms of the top5 accuracy. Furthermore, our gain of 1% (top1 accuracy) on ImageNet1k demonstrates that CAPro is robust to the domain gap between web and real-world datasets. Web data include advertisements, artworks, and renderings that differ from realistic photographs. On Google500 and ImageNet500, CAPro outperforms existing methods despite our disadvantages.

Table 2: Results on NUS-WIDE (Web).

| Method | Back-bone | NUS-WIDE C-F1 | O-F1 | mAP |
|---|---|---|---|---|
| Vanilla [78] | R50 | 37.5 | 39.6 | 43.9 |
| VSGraph [78] | R50 | 38.6 | 40.2 | 44.8 |
| MCPL [95] | R101 | 22.5 | 17.2 | 47.4 |
| Vanilla (ours) | R50 | 37.8 | 42.4 | 38.3 |
| CAPro (ours) | R50 | **39.3** | **45.4** | **48.0** |

Table 3: Results on open-set recognition.

| Method | WebVision C-F1 | ImageNet C-F1 |
|---|---|---|
| Vanilla [78] | 50.5 | 46.4 |
| CoTeach [20, 78] | 51.0 | 47.7 |
| VSGraph [78] | 57.2 | 52.8 |
| Vanilla (ours) | 54.6 | 48.3 |
| CAPro (ours) | **62.2** | **57.8** |

Table 2 reports per-Class F1 (C-F1), Overall F1 (O-F1), and mean Average Precision (mAP) on NUS-WIDE. Most prior multi-label methods are developed for ground-truth labels, while we are concerned with noisy WSL settings. Under this circumstance, CAPro is compared with methods that are trained on NUS-WIDE (Web) and evaluated on clean testing set. Following [96, 78], the top three categories of an image by prediction confidence are chosen for metric calculation. CAPro reaches the SOTA with a significant increase of 1.5% (C-F1), 3.0% (O-F1), and 9.7% (mAP) over our vanilla.

### 4.3  Discussion on Open-Set Recognition

To verify if CAPro can identify outliers of unknown categories, we conduct experiments on open-set recognition. Specifically, we train CAPro on Google500 and validate on the testing sets of WebVision1k and ImageNet1k. Images from the remaining 500 classes all belong to one open-set

category. We follow [97, 98, 78] to classify an image as open-set if its highest prediction confidence is below a threshold. The average C-F1 is adopted to reflect whether a model can discriminate between base and novel classes (501 in total). Table 3 confirms our superiority over existing methods, showing that CAPro is capable of detecting semantic novelty. More analysis can be found in Sec. E.

Table 4: Ablation study on text encoding, enhancement, and reference provider.

| Text Encoding | Text Enhancement | Reference Provider | Google500 Top1 | Top5 | ImageNet500 Top1 | Top5 | NUS-WIDE C-F1 | O-F1 | mAP |
|---|---|---|---|---|---|---|---|---|---|
| × | × | × | 71.5 | 87.8 | 66.5 | 84.6 | 37.2 | 42.4 | 46.2 |
| MiniLM | VSGraph [78] | × | 72.0 | 88.0 | 66.9 | 85.4 | 39.2 | 44.4 | 46.8 |
| MiniLM | ✓ (ours) | × | 75.5 | 91.0 | 71.5 | 88.8 | 39.3 | 44.9 | 47.4 |
| XLNet | VSGraph [78] | × | 71.6 | 87.8 | 66.8 | 84.8 | 38.6 | 43.4 | 47.6 |
| XLNet | ✓ (ours) | × | 75.4 | 91.0 | 71.5 | 88.8 | 39.3 | 45.1 | 47.5 |
| GPT-Neo | VSGraph [78] | × | 72.0 | 88.0 | 67.2 | 85.3 | 39.2 | 45.0 | 47.4 |
| GPT-Neo | ✓ (ours) | × | 75.7 | 91.1 | 71.6 | 88.8 | 39.2 | 45.1 | 47.6 |
| MiniLM | ✓ (ours) | Mix-up (MU) [99] | 75.7 | 90.9 | 71.4 | 88.6 | 38.7 | 45.3 | 47.2 |
| MiniLM | ✓ (ours) | Bootstrap [33] | 75.5 | 90.8 | 71.3 | 88.4 | 38.1 | 43.2 | 46.0 |
| MiniLM | ✓ (ours) | Label smooth [100] | 75.4 | 90.8 | 71.2 | 88.4 | 36.9 | 42.1 | 46.8 |
| MiniLM | ✓ (ours) | SCC [22] | 73.8 | 89.9 | 70.2 | 88.0 | 35.6 | 41.3 | 45.0 |
| MiniLM | ✓ (ours) | NCR [58] | 75.5 | 91.1 | 71.5 | 88.8 | 37.6 | 43.4 | 46.8 |
| MiniLM | ✓ (ours) | ✓ CB (ours) | 76.0 | 91.3 | 72.0 | 89.2 | 39.3 | 45.4 | 48.0 |
| MiniLM | ✓ (ours) | ✓ CB (ours) + MU | 76.5 | 91.1 | 71.9 | 88.8 | 40.4 | **46.7** | 49.9 |
| GPT-Neo | ✓ (ours) | ✓ CB (ours) | 76.1 | **91.4** | **72.1** | **89.4** | 39.3 | 44.9 | 47.7 |
| GPT-Neo | ✓ (ours) | ✓ CB (ours) + MU | **76.5** | 91.2 | 72.0 | 88.8 | **40.7** | 45.2 | **50.0** |

## 4.4 Ablation Study

**Text Encoding and Enhancement**    Table 4 reveals the benefit from cross-modality alignment. For the method without text encoding and enhancement, we sample $K$ examples randomly from each category. These instances barely provide reliable prototypes with semantic correctness. With respect to the text encoder, we additionally validate XLNet (base) [29] and GPT-Neo (1.3B) [31]. MiniLM surpasses XLNet by a minor margin, but both exhibit similar performance with our enhancement. GPT-Neo displays its power even with the plain $k$-NN-based smoothing in VSGraph [78], implying that advanced a Large Language Model (LLM) would boost CAPro. Note that encoders in CLIP [76] are not applicable here to avoid visual data leakage. All methods with text encoding and enhancement outperforms the vanilla one, validating the guidance from textual knowledge. Besides, we notice an increase up to 3.8%/4.7% on Google500/ImageNet500 by our text enhancement, showing that proper noise suppression is indispensable. Fig. 3 presents a comparison on the top-$K$ matched instances. Enhancement in VSGraph can filter out OOD to a certain degree, but can do nothing with lexical confusion (*e.g.*, food in *Drumstick* and players in *Tiger cat*). More qualitative results are in Sec. D.1.

**Reference Provider**    Our collective bootstrapping is compared against commonly used regularization methods which provide reference on targets. Surprisingly, most existing methods bring about no or even negative impact. In one respect, bootstrapping and label smoothing are already implicitly inherited in our framework as Eqs. (6) and (7). Therefore, no further gains can be seized. As for SCC, its estimated confidence may not comply with our Eq. (6), which leads to incompatible optimization. NCR has two drawbacks: 1) The chance of similar instances appear in one mini-batch is fairly small for large web datasets. 2) It only counts on self-prediction as reference source which is fragile to noise. In contrast, CAPro is enlightened by MoCo to maintain the dictionay as a queue, which enlarges the number of reference objects beyond instances in a mini-batch. Our enriched sources from both self-prediction and instance-prototype similarity expedite a steady learning progress. Moreover,  mix-up improves CAPro in top1 but lowers top5. It adopts convex combinations for both inputs and targets, enforcing a stronger regularization than our CB where we only recombine targets. For WebVision1k, examples with noisy labels still resemble prototypes and therefore neighbor knowledge brings useful reference. Mix-up does not consider appearance similarity and causes over-regularization.

**$\lambda^{\text{bts}}$ and top-$K$**    Fig. 4 shows that an increasing $\lambda^{\text{bts}}$ triggers off worse results on Google500 and ImageNet500. This suggests that collective knowledge should not overwhelm individual instance decision. With regard to top-$K$ on prototype initialization, there exists a trade-off between domain-variety and class-purity. The rise of $K$ increases diversity but also the risk of noise.

More ablation studies on the **threshold** $\gamma$, the **update frequency** of visual prototypes, and the **noise removal policy** can be found in Secs. D.2, D.3, and D.4, respectively. Empirical guidelines on **hyper-parameter tuning** are concluded in Sec. F. We also provide analysis of **failure cases** in Sec. G. Our computation cost with respect to performance gains can be found in Sec. H.

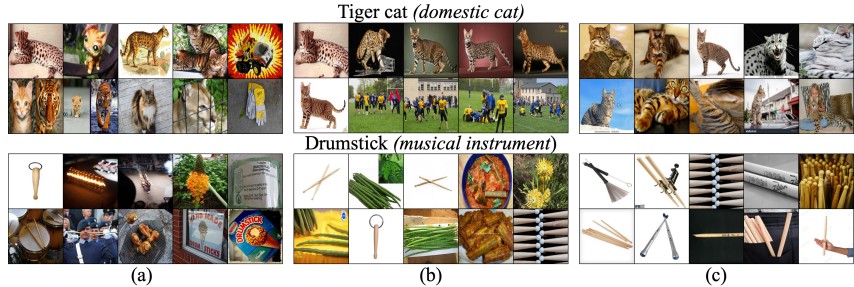

Figure 3: Top-matched WebVision1k instances are chosen: (a) without text enhancement, (b) with text enhancement in VSGraph [78], and (c) with our text enhancement.

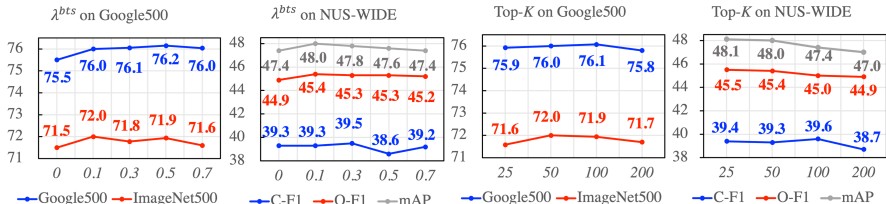

Figure 4: Impact of hyper-parameters $\lambda^{\text{bts}}$ and top-$K$ on CAPro.

## 5   Conclusion

CAPro utilizes web datasets to learn visual representations that are aligned with correct semantics. Cross-modality alignment and collective bootstrapping are corroborated as two keys to improve WSL. The benefits of building prototypes are two-fold: 1) Noisy web data whose visual and textual descriptions can be efficiently removed by simply measuring the distance between an instance and its prototype in the embedding space. 2) The inter-class relationship can be statistically studied by comparing each instance to all class prototypes, which may shed light on visual similarity for species. Three potential drawbacks should be considered: 1) the limited intra-class diversity with less tolerance to the minority in one category. Images crawled from websites follow the long-tailed distribution, which means that the more common or typical one instance is, the greater the likelihood that it gets exposed online. Over-emphasis on the purity of class prototypes leads to false negatives on the recognition of atypical samples. One possible solution is to introduce randomness into the initialization and update of prototypes to improve generalization. 2) the noteworthy domain gap and bias of web data. Even image contents are correct, their styles (*e.g.*, advertising photos, artworks, and rendering) are different from the realistic datasets. When it comes to modalities such as infrared or medical tomography images, there exist very few images online. Therefore, it is encouraged to prepare realistic images for guided-training and evaluation, where early-stopping and normalization techniques can be used to avoid overfitting. 3) the accuracy of prior knowledge about class hierarchy. Since definitions of class prototypes rely on the systematic understanding of concepts, improper, coarse-grained, or even wrong descriptions would devalue the semantic alignment. A thorough analysis on class concepts is a requisite to developing prototypes. Future work includes extension to other modalities (*e.g.*, audio and video clips) and to other tasks (*e.g.*, semi-supervised learning).

**Broader Impact**   CAPro manoeuvres language models for visual concept learning, where LLMs (*e.g.*, GPT-Neo) can deliver promising results for future development. Regardless of data sources, we showcase a cost-efficient and practical way to utilize cross-modality data. It also promotes rethinking the key-value matching mechanism for creative usages. For example, the visual dictionary originally built for instance-wise contrastive learning is re-discovered for our collective bootstrapping.

## Acknowledgements

The authors would like to express gratitude to the anonymous reviewers who greatly help improve the manuscript. In addition, the authors thank Hao Cheng from the University of California, Santa Cruz for the valuable comments.

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

# A Datasets Details

## A.1 WebVision1k

It contains 2.4M web images collected from Google and Flickr, which share the same 1k category names with ImageNet1k [1]. For each example, we use all available description, title, and tag in its metadata for raw text preparation. Besides, we follow [22, 78] to use the subset of WebVision–**Google500** for ablation studies in consideration of lower GPU resource and time consumption without losing generalization. It contains 0.48M images from Google with randomly chosen 500 categories. The testing set of ImageNet1k and its subset ImageNet500 are involved as well for evaluation.

## A.2 NUS-WIDE (Web)

It contains 0.26M web images from Flickr with 5k unique user tags. Each example is manually annotated with multiple labels within 81 concepts that are filtered out of the 5k tags. It also provides weak labels (an official web version) by checking if each of the 81 category name appears in user tags of every example. Almost 50% of the web labels are wrong and 50% of the true labels are missing in tags [88]. Since tags contain phrases without delimiters, we split phrases based on unigram frequencies [101] for raw text preparation. We follow [78] to train models with weakly-labeled training set (*a.k.a.*, NUS-WIDE Web) and validate them on the clean testing set.

# B Mathmatical Notations

In this section, we present the description of all math notations in the manuscript (see Table 5).

# C Implementation and Training Details

## C.1 General Settings

In consideration of performance and efficiency, we adopt ResNet-50 [89] and MiniLM-L6 [30] as image and text encoders by default. The training settings are listed in Table 6. In general, we refer to [28] for: batch size is 256; optimizer is SGD; learning rate linearly rises with 10 warm-up epochs and decays by cosine schedule. Specifically, although the benefit of an increased batch size (*e.g.*, 1024) has been validated [78, 58], we follow the standard batch size setting of 256 on WebVision1k for two reasons: 1) comparability with most of the previous methods; 2) limited computing resources with 8 GPUs (a mini-batch size of 32 on each GPU for a batch size of 256 in total).

We empirically set $\lambda^{prj} = 1$, $\lambda^{pro} = 1$, $\lambda^{ins} = 1$, $\lambda^{bts} = 0.1$, $m_p = 0.999$, $d_p = 128$, $\tau = 0.1$, $\alpha = 0.5$, and $K = 50$ by default. Their optimal values require meticulous fine-tuning of each hyper-parameter, which is beyond consideration of the present study.

Data augmentation (random cropping, resacling, and horizontal flipping) is applied on the inputs to the query encoder while stronger augmentation, including color jittering and blurring [50]), is added on those to the key encoder.

Experiments are conducted on a CentOS 7 workstation with an Intel 8255C CPU, 377 GB Mem, and 8 NVIDIA V100 GPUs. The training of CAPro on WebVision1k, Google500, and NUS-WIDE respectively costs about ten days, three days, and one day under the environment mentioned above.

## C.2 WebVision1k-only Implementation

We refer to [28] for $Q = 8192$ and $\gamma = 0.6$. The learning rate is 0.1 and the number of epochs is 120. Given the complexity of graph construction, we use $k = 5$ for both text denoising and matching.

## C.3 NUS-WIDE (Web)-only Implementation

In view of the dataset scale, we set $Q = 2048$ with a learning rate of $2 \times 10^{-3}$ and a total of 100 epochs. $k = 10$ is chosen since user tags are noiser and sparser in NUS-WIDE.

Table 5: List of symbols.

| Symbol | Description |
| --- | --- |
| $\mathbf{x}_i$ | a web image indexed with $i$ |
| $\mathbf{t}_i$ | textual metadata associated with $\mathbf{x}_i$ |
| $y_i$ | web label associated with $\mathbf{x}_i$ |
| $y_i^*$ | ground-truth label associated with $\mathbf{x}_i$ |
| $N$ | the total number of images in a web dataset |
| $C$ | the total number of categories |
| $D$ | a web dataset |
| $\theta_e$ | parameters of image encoder |
| $\theta_c$ | parameters of classifier |
| $\mathcal{F}(\theta_e; \theta_c)$ | a deep model with image encoder and classifier |
| $d_v$ | dimension of the visual features |
| $\mathbf{v}_i$ | visual features of $\mathbf{x}_i$ extracted by image encoder, $\mathbf{v}_i \in \mathbb{R}^{d_v}$ |
| $d_t$ | dimension of the textual embeddings |
| $\mathbf{s}_i$ | textual embeddings of $\mathbf{t}_i$ extracted by text encoder, $\mathbf{s}_i \in \mathbb{R}^{d_t}$ |
| $\mathbf{t}^c$ | category definition of the $c$-th class |
| $\mathbf{s}^c$ | textual embeddings of $\mathbf{t}^c$ extracted by text encoder, $\mathbf{s}^c \in \mathbb{R}^{d_t}$ |
| $\mathbf{p}_i$ | predictions on $\mathbf{v}_i$ from classifier, $\mathbf{p}_i \in \mathbb{R}^C$, $\mathbf{p}_{i(k)}$ denotes its $k$-th element |
| $d_p$ | dimension of the visual embeddings |
| $\mathbf{z}_i$ | low-dimensional embeddings of $\mathbf{v}_i$ after projection, $\mathbf{z}_i \in \mathbb{R}^{d_p}$ |
| $\tilde{\mathbf{v}}_i$ | reconstructed visual features from $\mathbf{z}_i$, $\tilde{\mathbf{v}}_i \in \mathbb{R}^{d_v}$ |
| $\mathbf{q}_i$ | predictions on $\mathbf{z}_i$ from auxiliary classifier, $\mathbf{q}_i \in \mathbb{R}^C$ |
| $Q$ | the size of dictionary, namely the length of queue |
| $k$ | number of nearest neighbors (NN) in $k$-NN and $k$-reciprocal-NN |
| $\mathcal{V}$ | vertices, nodes |
| $\mathcal{E}$ | edges |
| $\mathcal{G}$ | graph, $\mathcal{G} = \{\mathcal{V}, \mathcal{E}\}$ |
| $A$ | adjacency matrix |
| $\mathcal{N}(\mathbf{x}_i, k)$ | $k$-NN sets of $\mathbf{x}_i$ |
| $\mathcal{R}(\mathbf{x}_i, k)$ | $k$-reciprocal-NN sets of $\mathbf{x}_i$ |
| $d(\mathbf{v}_i, \mathbf{v}_j)$ | distance between $\mathbf{v}_i$ and $\mathbf{v}_j$ |
| $d^*(\mathbf{v}_i, \mathbf{v}_j)$ | refined distance between $\mathbf{v}_i$ and $\mathbf{v}_j$ |
| $V_{\mathbf{v}_i, \mathbf{v}_j}$ | $k$-reciprocal feature encoding the distance between $\mathbf{v}_i$ and $\mathbf{v}_j$ |
| $\mathbf{S}$ | concatenated textual embeddings, $\mathbf{S} = (\mathbf{s}_1, \mathbf{s}_2, ..., \mathbf{s}_N), \mathbf{S} \in \mathbb{R}^{N \times d_t}$ |
| $I_N$ | identity matrix |
| $\tilde{A}$ | adjacency matrix $A$ with self-connection $I_N$ |
| $\tilde{D}_{ii}$ | degree matrix of $\tilde{A}$ |
| $\hat{\mathbf{S}}$ | denoised $\mathbf{S}$ after smoothing |
| $\hat{\mathbf{s}}_i$ | denoised $\mathbf{s}_i$ after smoothing |
| $K$ | top-$K$ selected examples with visual-semantic alignment |
| $\sigma_K^c$ | the $K$-th smallest distance $d^*(\hat{\mathbf{s}}_i, \mathbf{s}^c)$ in the $c$-th class |
| $D_K^c$ | top-$K$ set of the $c$-th class |
| $D_K$ | sets of top-$K$ examples from all classes |
| $\hat{\mathbf{z}}^c$ | unnormalized visual prototype of the $c$-th class |
| $\mathbf{z}^c$ | normalized visual prototype of the $c$-th class |
| $\tau$ | temperature coefficient |
| $\alpha$ | weight between self-prediction and prototype-instance similarity |
| $\mathbf{o}_i$ | fused, comprehensive output for label correction, $\mathbf{o}_i \in \mathbb{R}^C$ |
| $\mathbf{r}_i$ | prototype-instance similarity, $\mathbf{r}_i \in \mathbb{R}^C$, $\mathbf{r}_{i(k)}$ denotes its $k$-th element |
| $\gamma$ | threshold for label correction |
| $m_p$ | momentum coefficient |
| $\mathbf{b}_i$ | collective bootstrapping target on $\mathbf{z}_i$ |
| $w_{ij}$ | weight of contribution from $\mathbf{z}'_j$ in the dictionary for $\mathbf{b}_i$ |
| $\lambda^{bts}$ | weight for collective bootstrapping loss |
| $\lambda^{prj}$ | weight for projection and reconstruction losses |
| $\lambda^{pro}$ | weight for prototypical contrastive loss |
| $\lambda^{ins}$ | weight for instance contrastive loss |

Table 6: List of hyper-parameters for training settings.

| Settings | WebVision1k/Google500 | NUS-WIDE |
|---|---|---|
| Optimizer | SGD | |
| Optimizer momentum | 0.9 | |
| Optimizer weight decay | $1 \times 10^{-4}$ | |
| Batch size | 256 | |
| Step1 pre-training scheduler | Cosine decay with linear warm-up | |
| Step1 pre-training warm-up epochs | 5/10 | 10 |
| Step1 pre-training learning rate | $1 \times 10^{-1}$ | $2 \times 10^{-3}$ |
| Step1 pre-training epochs with encoders frozen | 0 | |
| Step1 pre-training epochs in total | 120 | 100 |
| Step2 training scheduler | Cosine decay with linear warm-up | |
| Step2 training warm-up epochs | 5/10 | 10 |
| Step2 training learning rate | $1 \times 10^{-1}$ | $2 \times 10^{-3}$ |
| Step2 training epochs with encoders frozen | 5/10 | 10 |
| Step2 training epochs in total | 60/120 | 100 |
| Step3 fine-tuning scheduler | Cosine decay | |
| Step3 fine-tuning warm-up epochs | 0 | |
| Step3 fine-tuning learning rate | $1 \times 10^{-4}$ | $2 \times 10^{-5}$ |
| Step3 fine-tuning epochs with encoders frozen | 15/20 | 20 |
| Step3 fine-tuning epochs in total | 15/20 | 20 |
| Image encoder (by default) | ResNet-50 [89] | |
| Text encoder (by default) | MiniLM-L6 [30] | |
| $\lambda^{prj}$ | 1 | |
| $\lambda^{pro}$ | 1 | |
| $\lambda^{ins}$ | 1 | |
| $\lambda^{bts}$ | 0.1 | |
| $m_p$ | 0.999 | |
| $d_p$ | 128 | |
| $\tau$ | 0.1 | |
| $\alpha$ | 0.5 | |
| top-$K$ | 50 | |
| Q | 8192 | 2048 |
| $\gamma$ | 0.6 | 0.9 |
| $k$-NN/$k$-reciprocal-NN | 5 | 10 |

To support multi-label learning, it is necessary, but not yet enough, to simply replace the softmax-based cross-entropy losses with sigmoid-based binary cross-entropy losses. The premise of prototypical contrastive learning does not hold true anymore because one instance can be simultaneously engaged in formation of multiple clusters, which violates the exclusivity inherited in softmax activation. Our experiments demonstrate that, only by projecting $\mathbf{v}_i$ into compact subspaces specific to each class, can we properly learn the decision boundary to continue prototypical learning. Technically, we set $C$ additional fully-connected (FC) layers after the projector to respectively map $\mathbf{z}_i$ into $\tilde{\mathbf{z}}_{i,c} \in \mathbb{R}^{d_p}, c = 1, 2, ..., C$. For the $c$-th class, both positive and negative prototypes $\tilde{\mathbf{z}}^{c+}, \tilde{\mathbf{z}}^{c-}$ are shaped accordingly for the contrast against $\tilde{\mathbf{z}}_{i,c}$. Such operation can be viewed as magnifying class-indicative contents via recombination of the shared embedding $\mathbf{z}_i$. Another minor modification is required on noise removal, where the output $\mathbf{o}_{i,c}$ is fused independently for the $c$-th class for binary separation. Considering the overwhelming negative instances, we set $\gamma = 0.9$ to avoid deviation by majorities in label rectification. Discussion on the hyper-parameter $\gamma$ can be found in Sec. D.2.

## C.4 Training Steps

CAPro adopts a three-step training pipeline. It employs a pre-training step to learn common visual patterns from the original web dataset $D$. Then, the training starts with instance-prototype and instance-wise contrastive learning, collective bootstrapping, and on-line noise removal. Finally, given the trained model, off-line noise removal is performed on $D$ for data cleaning and we fine-tune the classifier alone with the cleaned dataset.

**Step1 pre-training** We perform visual pre-training on ResNet-50 with cross-entropy and projection-reconstruction losses. At this time, we only use original web labels to train the model to learn common visual description, which lays foundation for the subsequent prototype initialization in step2. Note that the **pre-trained** model is also the **vanilla** method in our experiments.

**Step2 training** All components are initialized with the pre-trained parameters in step1. As shown in Table 6, we keep the encoders frozen with warm-up epochs. This helps stabilize training to avoid the prototypical embeddings, which are initialized at the beginning by averaging top-$K$ semantically-correct examples, being perturbed drastically. In this step, we re-train the model with all losses. Apart from classification, we perform instance-wise and instance-prototype contrastive learning, collective bootstrapping, noise removal, and prototype update.

**Step3 fine-tuning** We follow MoPro [28] to perform noise removal on the training dataset $D$. The trained model in step2 is used to correct labels or discard OOD examples with our control flow. Then, we keep encoders frozen and fine-tune the classifier alone with the cleaned set for better performance. Note that such **fine-tuned** model is exactly the **CAPro** in experiments.

---

**Algorithm 1:** CAPro's training procedure.

---

**Data:** Web images and their associated texts and labels $D = \{(\mathbf{x}_i, \mathbf{t}_i, y_i)\}_{i=1}^N$.

1 **Step1 pre-training**
2 **for** $(\mathbf{x}_i, y_i) \in D$ **do**
3     $\mathcal{L}_i = \mathcal{L}_i^{\mathrm{cls}} + \mathcal{L}_i^{\mathrm{prj}}$;
4     Update image encoder, classifier, projector, and reconstructor to minimize $\mathcal{L}_i$;
5 **end**
6 **Step2 training**
7 **for** $(\mathbf{x}_i, \mathbf{t}_i, y_i) \in D$ **do**
8     Extract $\mathbf{v}_i$ from $\mathbf{x}_i$ via the image encoder;
9     Extract $\mathbf{s}_i$ from $\mathbf{t}_i$ via the text encoder;
10 **end**
11 Build $k$-reciprocal-NN graph $\mathcal{G} = \{\mathcal{V}, \mathcal{E}\}$ with $\{\mathbf{v}_i\}_{i=1}^N$;
12 Enhance text embeddings from $\mathbf{s}_i$ to $\hat{\mathbf{s}}_i$ via graph convolution on $\mathcal{G}$;
13 **for** $c \in \{1, 2, ..., C\}$ **do**
14     Extract $\mathbf{s}^c$ from $\mathbf{t}^c$ via the text encoder;
15     **for** $i \in \{1, 2, ..., N | y_i = c\}$ **do**
16        Match textual instances $\mathbf{s}_i$ to prototypes $\mathbf{s}^c$ to obtain visual anchors $D_K^c$;
17     **end**
18 **end**
19 Initialize visual prototypes with $D_K$;
20 **for** $(\mathbf{x}_i, y_i) \in D$ **do**
21     $\mathcal{L}_i = (1 - \lambda^{\mathrm{bts}})\mathcal{L}_i^{\mathrm{cls}} + \lambda^{\mathrm{bts}}\mathcal{L}_i^{\mathrm{bts}} + \lambda^{\mathrm{prj}}\mathcal{L}_i^{\mathrm{prj}} + \lambda^{\mathrm{pro}}\mathcal{L}_i^{\mathrm{pro}} + \lambda^{\mathrm{ins}}\mathcal{L}_i^{\mathrm{ins}}$;
22     Update image encoder, classifier, and projector to minimize $\mathcal{L}_i$;
23     Refine $y_i$ to $\hat{y}_i$ to remove noise;
24 **end**
25 **Step3 fine-tuning**
26 **for** $(\mathbf{x}_i, \hat{y}_i) \in D$ **do**
27     $\mathcal{L}_i = \mathcal{L}_i^{\mathrm{cls}}$;
28     Update classifier to minimize $\mathcal{L}_i$;
29 **end**

---

We also prepare Algo. 1 to explicitly explain the entire training process.

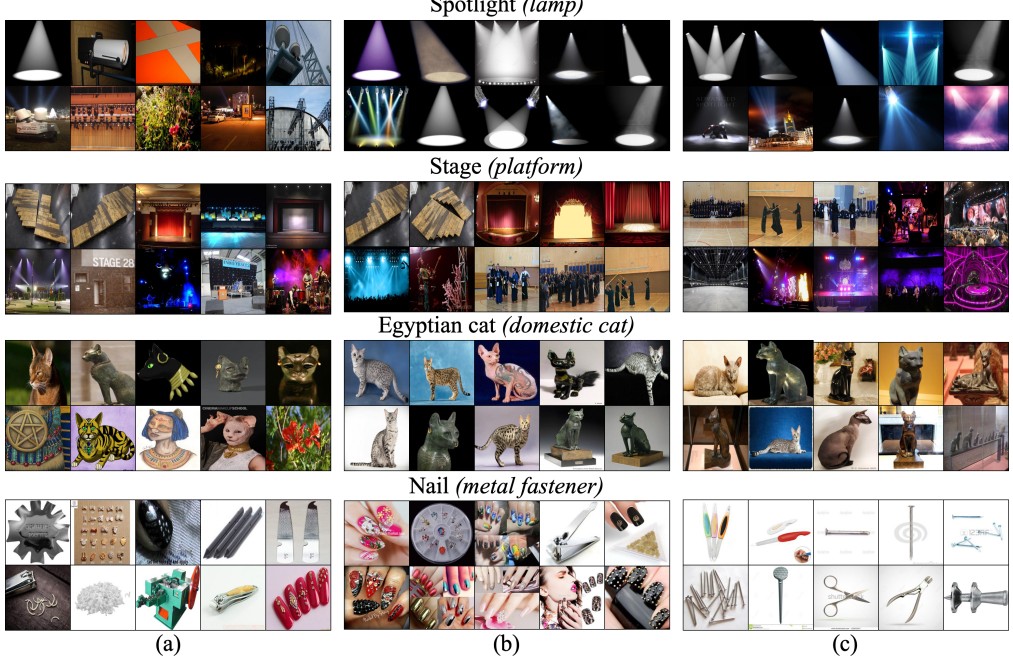

Figure 5: Top-matched WebVision1k instances are chosen: (a) without text enhancement, (b) with text enhancement in VSGraph [78], and (c) with our text enhancement.

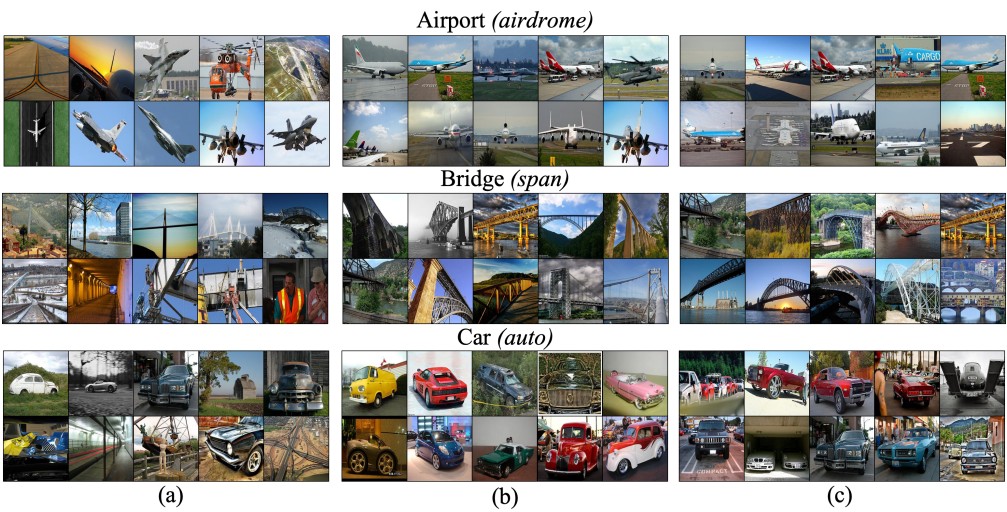

Figure 6: Top-matched NUS-WIDE (Web) instances are chosen: (a) without text enhancement, (b) with text enhancement in VSGraph [78], and (c) with our text enhancement.

# D  Ablation Study

## D.1  Effect of Text Enhancement

Figs. 5 and 6 present additional qualitative comparison for selecting instances with potentially-correct semantics. For WebVision, noiser categories are chosen to validate the effectiveness of text enhancement by smoothing and reranking. We can observe that due to the problem of polysemy, a majority of the retrieved images are irrelevant to the correct semantics and simple $k$-NN-based smoothing in [78] can hardly handle such situation. In contrast, our text enhancement helps pinpoint, not perfect but comparatively reliable, web instances that share similar semantics (*e.g.*, metalwork in *Nail*). Besides, we also sample three categories from the noiser NUS-WIDE to double-check the effectiveness of text enhancement. For example, in the category of airport, direct matching of user tag embeddings to the textual prototype returns a few close-up images of warcrafts, which has nothing to do with *Airport*. On the contrary, our text enhancement helps to select the truly matched instances.

Table 7: Effect of $\gamma$ on CAPro without collective bootstrapping.

| $\gamma$ | Reference Provider | Google500 | | ImageNet500 | | NUS-WIDE | | |
|---|---|---|---|---|---|---|---|---|
| | | Top1 | Top5 | Top1 | Top5 | C-F1 | O-F1 | mAP |
| 0.6 | $\times$ | 72.0 | 88.0 | 66.9 | 85.4 | 8.3 | 9.1 | 6.9 |
| 0.8 | $\times$ | 71.2 | 87.7 | 65.9 | 84.8 | – | – | – |
| 0.9 | $\times$ | – | – | – | – | 39.2 | 44.4 | 46.8 |

## D.2  Effect of $\gamma$ on Noise Removal

Table 7 reports the influence of $\gamma$ when collective bootstrapping is not applied. We follow MoPro [28] to validate two $\gamma$ candidates: $\gamma = 0.6$ and $\gamma = 0.8$. We find that $\gamma = 0.6$ works the best on Google500. As $\gamma$ increases, it becomes more difficult for the model to correct labels and thereafter label-flipping errors may still exist. As suggested by MoPro, the optimum $\gamma$ is related to the percentage of noise in web datasets. For noiser datasets, $\gamma$ should be decreased for the model to correct wrong labels at an early stage. Otherwise, overfitting might occur and weaken prototypical representation learning. The fine-tuning of $\gamma$ requires elaborate experiments, which is beyond the scope of the present study.

For multi-label learning on NUS-WIDE, the optimum $\gamma$ is not only related to the noise level but also to the ratio of the number of positive examples to that of negative examples. Since the negative instances in each class exceeds positive ones by more than one order of magnitude, decreasing $\gamma$ will easily make the model to classify any instance as negative. Once the model overfits the overwhelming negative examples, valid positive supervision sources would only come from the top-$K$ matched examples with cross-modality alignment, which degrades generalizability greatly. In this case, we should keep a stricter threshold $\gamma = 0.9$ to only allow confident label rectification.

Table 8: Effect of prototype update frequency on CAPro. By default, we update visual prototypes every epoch using high-quality examples in each mini-batch. For 0-epoch per update, we do not introduce additional high-quality web examples to polish prototypes, but only update them with the top-$K$ matched semantically-correct examples with their latest visual embeddings.

| # Epochs per update | Google500 | | ImageNet500 | | NUS-WIDE | | |
|---|---|---|---|---|---|---|---|
| | Top1 | Top5 | Top1 | Top5 | C-F1 | O-F1 | mAP |
| 0 | 75.5 | 91.1 | 71.6 | 88.8 | 39.2 | 44.4 | 47.2 |
| 1 (by default) | 76.0 | 91.3 | 72.0 | 89.2 | 39.3 | 45.4 | 48.0 |
| 5 | 75.9 | 91.2 | 71.8 | 89.2 | 39.6 | 45.0 | 47.6 |
| 10 | 76.0 | 91.2 | 71.7 | 89.1 | 39.3 | 45.8 | 48.2 |

## D.3  Effect of Prototype Update Frequency

By default, we update prototypes by examples in a mini-batch for every epoch. We additionally perform ablation study on the update frequency where comparison is conducted between: 1) 0-epoch, 2) 1-epoch (by default), 3) 5-epoch, and 4) 10-epoch. For 0-epoch update, we **do not** update prototypes with embeddings from other high-quality web examples. Instead, prototypes are **renewed**

with the latest embeddings only from the top-$K$ matched examples to avoid being out-of-date. For 5-epoch and 10-epoch update, we polish prototypes every 5 and 10 epochs, respectively.

Table 8 reports the effect of prototype update frequency on CAPro. On the Google500 and NUS-WIDE datasets, if prototypes are only formed by limited clean examples, their generalization across domain becomes poor and thus causes performance drop of 0.45% (top1) and 0.6% on average, respectively. For Google500, reduced update frequency generally causes lower performance, meaning that the prototypes should keep refreshed for large-scale datasets. For NUS-WIDE, if the update frequency decreases a bit (5-epoch), the model improves on C-F1 but underperforms a little on O-F1 and mAP. With the 10-epoch frequency, we surprisingly find that results are improved on all evaluation metrics. One possible explanation is that the delayed prototype update can help stabilize training at an early stage, but the optimal frequency might be subject to dataset scale and noise level.

Table 9: Effect of noise removal policy on CAPro. We compare with MoPro to show the effectiveness of keeping labels of top-$K$ matched semantically-correct examples unchanged.

| Noise Removal | Google500 | | ImageNet500 | | NUS-WIDE | | |
|---|---|---|---|---|---|---|---|
| policy | Top1 | Top5 | Top1 | Top5 | C-F1 | O-F1 | mAP |
| MoPro [28] | 75.8 | 91.1 | 71.7 | 89.0 | 38.8 | 42.2 | 47.2 |
| CAPro (ours) | 76.0 | 91.3 | 72.0 | 89.2 | 39.3 | 45.4 | 48.0 |

### D.4 Effect of Noise Removal Policy

Table 9 reports the effect of noise removal policy on CAPro. Our label adjustment policy is inspired from the Eq. (5) of MoPro [28] but differs in that we keep labels of the top-$K$ matched examples selected in cross-modality alignment unchanged. Therefore, if we replace our noise removal policy with the MoPro one, we actually allow the deep model to get rid of the guidance from top-$K$ examples. These selected top-$K$ examples can be altered or even discarded by the MoPro-style policy.

It turns out that without such enforcement, the performance dropped under both single-label and multi-label scenarios. The possible reason behind is that, due to the overwhelming noise (*e.g.*, the semantic-misalignment noise) in certain categories, the model itself cannot keep representation learning robust to noise even with a good start. The labels of top-$K$ samples will be prone to the noisy majority, which invalidates prototypical learning. Besides, the superiority of CAPro over MoPro-style update also substantiates the purity and correctness of the selected examples.

## E Analysis on Open-Set Recognition

We provide detailed analysis on CAPro for open-set recognition. Fig. 7 presents the per-class F1 (C-F1), per-class Precision (C-P), and per-class Recall (C-R). We compare five methods for ablation study including: 1) the vanilla method, 2) CAPro without text enhancement (TE) & collective bootstrapping (CB), 3) CAPro without CB, 4) CAPro, and 5) CAPro with mix-up (MU).

We vary the confidence threshold from 0 to 1 with an interval of 0.01 to comprehensively measure the performance of CAPro. For each example, if the highest model prediction confidence is below the threshold, the example will be classified as the open-set category. Otherwise, the example will be classified into one of the known categories. We train methods on the Google500 training set and validate them on WebVision1k and ImageNet1k testing sets. Examples from the remaining 500 novel categories should all fall into the open-set category.

It can be observed that compared with vanilla method, CAPro enjoys a much higher recall but a relatively lower precision. It means that our CAPro is more confident about its prediction and a threshold around 0.6 would end up with an optimal C-F1 over 66%. The precision-recall curve reflects that each key component does improve open-set recognition. Note that due to the limited sampling of confidence threshold, the precision-recall curve is not spanning across the entire axes. However, the tendency of curves confirms the effectiveness of our components.

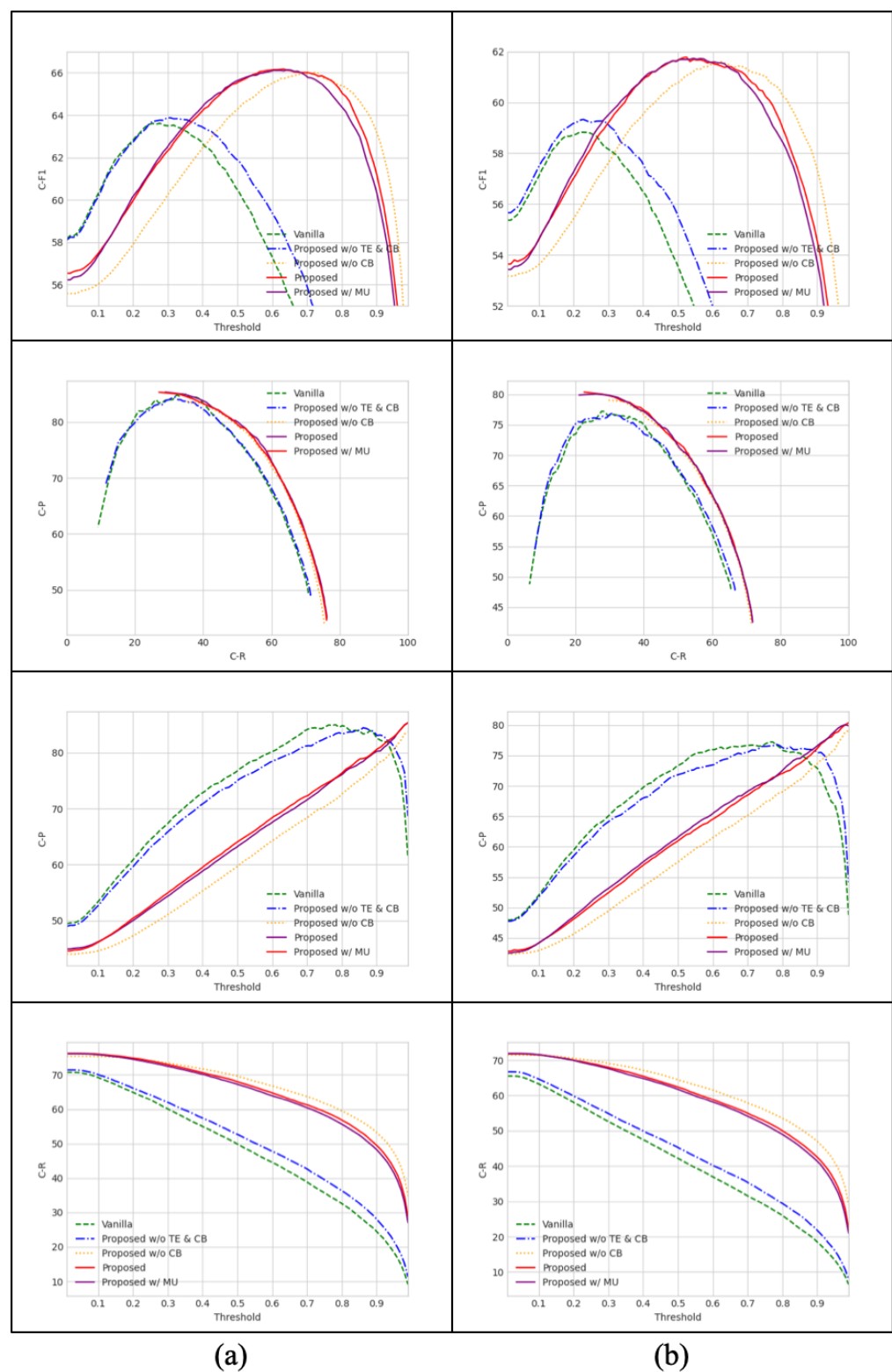

Figure 7: Effect of threshold for open-set recognition on (a) WebVision1k and (b) ImageNet1k. TE, CB, and MU respectively refer to our text enhancement, collective bootstrapping, and mix-up. Examples with prediction confidence lower than the threshold will be classified as open-set category.

# F  Guidelines on Tuning of Hyper-Parameters

**Loss weights of $\lambda^{\mathbf{bts}}$, $\lambda^{\mathbf{prj}}$, $\lambda^{\mathbf{pro}}$, and $\lambda^{\mathbf{ins}}$**    First, for the total objective, we follow MoPro [28] to use $\lambda^{\mathrm{pro}} = 1$ and $\lambda^{\mathrm{ins}} = 1$. Out of simplicity, we also use $\lambda^{\mathrm{prj}} = 1$ as default.

Second, we would like to explain the effect of $\lambda^{\mathrm{pro}}$, $\lambda^{\mathrm{ins}}$, and $\lambda^{\mathrm{prj}}$ on regularzation. A larger $\lambda^{\mathrm{pro}}$ may pull instances too close totheir prototypes, which "shrinks" class clusters in the embedding space. A larger $\lambda^{\mathrm{ins}}$ will enforce stronger visual discriminability between two instances. It may cause two examples from the same category to differ greatly and thereafter downgrades the visual prototype update and class cluster regularization. A larger $\lambda^{\mathrm{prj}}$ improves the reconstruction quality of $\tilde{\mathbf{v}}_i$, which encourages $\mathbf{z}_i$ to retain more information of $\mathbf{v}_i$ in the embedding space. The projection-reconstruction loss is only involved in the pre-training stage (see Algo. 1), and therefore $\lambda^{\mathrm{prj}}$ will not affect the prototypical and instance-wise contrastive learning in the following stage.

Third, for one's custom web datasets, we suggest that $\lambda^{\mathrm{pro}}$, $\lambda^{\mathrm{ins}}$, and $\lambda^{\mathrm{prj}}$ should be tuned according to the performance results under three settings: 1)$\lambda^{\mathrm{pro}} = 0$ vs. $\lambda^{\mathrm{pro}} = 1$; 2)$\lambda^{\mathrm{ins}} = 0$ vs. $\lambda^{\mathrm{ins}} = 1$; 3)$\lambda^{\mathrm{prj}} = 0$ vs. $\lambda^{\mathrm{prj}} = 1$.

According to our experiments on both single-label and multi-label datasets, the default settings of $\lambda^{\mathrm{pro}} = 1$, $\lambda^{\mathrm{ins}} = 1$, and $\lambda^{\mathrm{prj}} = 1$ should work well on most cases.

For $\lambda^{\mathrm{bts}}$, we suggest 0.1 would achieve a balance between the individual and collective label references. A much larger value may cause over smoothing and over-regularization on visual learning.

**Threshold $\gamma$**    For $\gamma$ on single-label datasets, its value is related to the percentage of noise in datasets. For WebVision1k and Google500 (34% noise [3]), $\gamma = 0.6$ works better than $\gamma = 0.8$. For one's own web dataset, if the noise ratio is larger, $\gamma$ should be tuned lower so that wrong labels could be corrected at an earlier stage before overfitting. For $\gamma$ on multi-label datasets, its value is related to both the percentage of noise and the number ratio of positive-negative samples. For NUS-WIDE (50% noise [78] and 0.02 avg. ratio of positive-negative examples), $\gamma = 0.9$ works better than $\gamma = 0.6$. For one's own web dataset, if the noise ratio is smaller and the positive over negative ratio is smaller, $\gamma$ should be tuned higher so that hard positive samples will not be easily discarded to avoid underfitting.

**Prototype Update Frequency**    For the update frequency, its value is related to the dataset scale and noise level. For WebVision1k and Google500, visual prototypes should be updated per epoch to improve their diversity, which better handles the domain gap between web and realistic datasets. For NUS-WIDE, the update frequency could be reduced to stabilize training, where the prototypes can be prone to the overwhelming negative examples in each category.

**Top-$K$**    For top-$K$, its value is related to the percentage of noise. If the noise ratio is less than 30%, $K$ should be set higher than 50 to include more diverse examples.

**Others**    The current settings of other hyper-parameters (see Table 6) work well. For one's own dataset, we believe these values can be set as starting points and finetuned accordingly. Among all hyper-parameters, our ablation results show that for $\lambda^{\mathrm{bts}}$, $\gamma$, and top-$K$, their values do affect performance and should be set following the rules mentioned above (such as the dataset scale, the noise ratio, and the positive-negative ratio). For the remaining hyper-parameters such as the prototype update frequency, we do not observe significant fluctuation. In other words, the model is robust to these hyper-parameters.

# G  Failure Cases

We provide failure cases of our CaPro on the WebVision1k dataset (see Fig. 8). Our observations are summarized as the following.

First, CAPro can handle fine-grained categories on WebVision1k. The introduction of atypicals increases the risk of noise. For generalization on anomalies or rarities, one solution is to choose both top-K and randomly sampled instances.

Second, for WebVision1k, both MoPro and CAPro underperform the vanilla baseline (optimized only by the cross-entropy loss) on a total of 387 and 312 classes, respectively. Top5 failures of classes

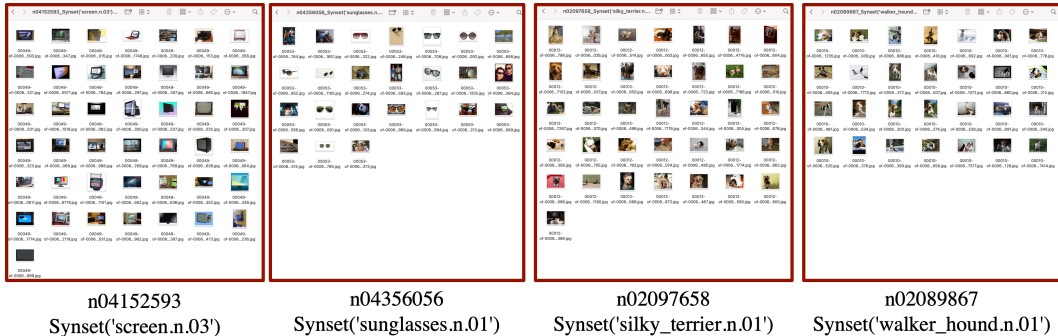

| n04152593 | n04356056 | n02097658 | n02089867 |
|:---:|:---:|:---:|:---:|
| Synset('screen.n.03') | Synset('sunglasses.n.01') | Synset('silky_terrier.n.01') | Synset('walker_hound.n.01') |

Figure 8: Failure cases of our CAPro on certain classes where the simple vanilla baseline achieves better performance.

include screen, sunGlasses, bellCote, ballPlayer, and popBottle. For ImageNet1k, MoPro and CAPro underperform the vanilla on a total of 450 and 358 classes, respectively. Top-5 failures of classes include silkyTerrier, walkerHound, academicGown, standardSchnauzer, and bellCote.

We also provide interesting findings below:

First, the domain gap exists between web and realistic datasets as the top 5 failure cases on the WebVision1k and ImageNet1k testing sets are quite different.

Second, the vanilla method tends to overfit the training set so that it outperforms on highly similar concepts such as screen vs. monitor and sunGlasses vs. sunGlass.

Third, mistakes on silky vs. yorkshire Terrier and walker vs. englishFox hound are ascribed to over-regularization. The inter-class relationship might be used for class-wise adjustment.

## H  Computational Complexity

Table 10: The parameters and GFLOPs of different encoders.

| Encoders | Number of Parameters | GFLOPs |
|---|---|---|
| R50 [89] | 25M | 3.8 |
| MiniLM [30] | 22M | 4.7 |
| XLNet [29] | 110M | 29 |
| GPT-Neo [31] | 1.3B | 3400 |

First, we present the number of parameters and GFLOPs for the image and text encoders in Table 11.

Table 11: The cost of the text enhancement of our CAPro with respect to its performance gains.

| Text Encoding | Text Enhancement | Cost | Google500 Top1 | ImageNet500 Top1 |
|---|---|---|---|---|
| MiniLM [30] | VSGraph [78] | $O(N^2 d_v)+O(N d_v{}^2 + Nk d_v)$ | 72.0 | 66.9 |
| | Ours | $+O(3Nk d_v)+O(4k)+O(4k log(4k))$ | +3.5 | +4.6 |
| XLNet [29] | VSGraph [78] | $O(N^2 d_v)+O(N d_v{}^2 + Nk d_v)$ | 71.6 | 66.8 |
| | Ours | $+O(3Nk d_v)+O(4k)+O(4k log(4k))$ | +3.8 | +4.7 |
| GPT-Neo [31] | VSGraph [78] | $O(N^2 d_v)+O(N d_v{}^2 + Nk d_v)$ | 72.0 | 67.2 |
| | Ours | $+O(3Nk d_v)+O(4k)+O(4k log(4k))$ | +3.7 | +4.4 |

Second, we present the cost of the text enhancement of our CAPro with respect to its performance gains. Here, $N$ denotes the number of all nodes in the visual graph. $k$ is the number of neighbors per node and $d_v$ is the dimension of the feature $\mathbf{v}$. With $k = 5$ and $k = 10$ respectively for WebVision1k and NUS-WIDE, our improvement over VSGraph is worthy at the expense of such a low cost.

Third, we present the cost of the reference provider of our CAPro with respect to its performance gains. Here, $m$ is the batch size, $d_v$ is the dimension of $\mathbf{v}$, $Q$ is the size of dictionary, $d_p$ is the

Table 12: The cost of the reference provider of our CAPro with respect to its performance gains.

| Text Encoding | Reference Provider | Cost | Google500 Top1 | ImageNet500 Top1 |
|---|---|---|---|---|
| MiniLM [30] | Mix-up [99] | – | 75.7 | 71.4 |
| | NCR [58] | $O(m^2(d_v + C))$ | -0.2 | +0.1 |
| | Our CB | $O(mQ(d_p + C))$ | +0.3 | +0.6 |
| MiniLM [30] | Bootstrap [33] | – | 75.5 | 71.3 |
| | NCR [58] | $O(m^2(d_v + C))$ | +0 | +0.2 |
| | Our CB | $O(mQ(d_p + C))$ | +0.5 | +0.7 |
| MiniLM [30] | LabelSmooth [100] | – | 75.4 | 71.2 |
| | NCR [58] | $O(m^2(d_v + C))$ | +0.1 | +0.3 |
| | Our CB | $O(mQ(d_p + C))$ | +0.6 | +0.8 |
| MiniLM [30] | SCC [22] | – | 73.8 | 70.2 |
| | NCR [58] | $O(m^2(d_v + C))$ | +1.7 | +1.3 |
| | Our CB | $O(mQ(d_p + C))$ | +2.2 | +1.8 |

dimension of $z$, and $C$ is the number of classes. The common reference provider techniques such as the Mix-up, Bootstrapping, label smoothing, and SCC do not incur significant overhead. Operations of our CB are fast to compute for moderate $m = 256$, $Q = 8192$, $d_p = 128$, and $C = 1000$ since PyTorch supports efficient matrix multiplication on GPUs. Besides, compared with NCR, our $d_p = 128$ is 16x smaller than $d_v$=2048 in NCR, and our $m = 256$ is 4x smaller than $m = 1024$ in NCR. For WebVision1k, our cost is 1.35x smaller than NCR. For NUS-WIDE, our cost is 20.37x smaller than NCR. It is reasonable to conclude that our CB is more efficient and effective than NCR.

Finally, it is noted that the text encoding and text enhancement methods are performed off-line and executed only once. They do not participate in network optimization. Besides, the pretrained text encoders are only used for inference under 1 V100 GPU. Therefore, the additional cost is acceptable in return for semantically-correct web images.

