# Supplementary Material

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

We also prepare Algo. S1 to explicitly explain the entire training process.

## S4 Ablation Study

### S4.1 Effect of Text Enhancement

Figs. S1 and S2 present additional qualitative comparison for selecting instances with potentially-correct semantics. For WebVision, noiser categories are chosen to validate the effectiveness of text enhancement by smoothing and reranking. We can observe that due to the problem of polysemy, a majority of the retrieved images are irrelevant to the correct semantics and simple $k$-NN-based smoothing in [3] can hardly handle such situation. In contrast, our text enhancement helps pinpoint, not perfect but comparatively reliable, web instances that share similar semantics (*e.g.*, metalwork in *Nail*). Besides, we also sample three categories from the noiser NUS-WIDE to double-check the effectiveness of text enhancement. For example, in the category of airport, direct matching of user tag embeddings to the textual prototype returns a few close-up images of warcrafts, which has nothing to do with *Airport*. On the contrary, our text enhancement helps to select the truly matched instances.

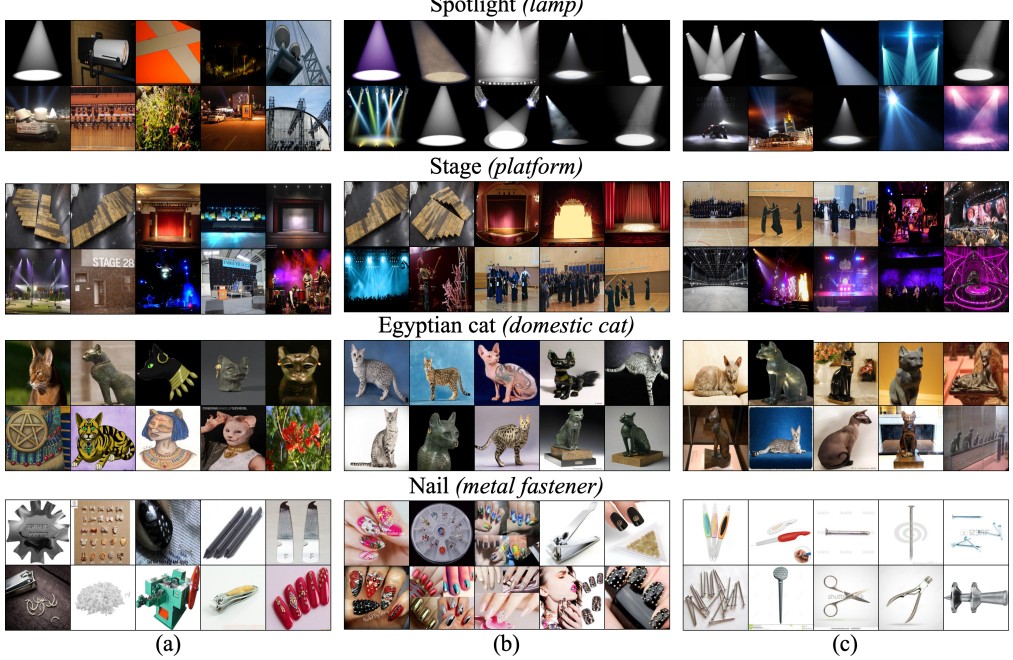

Figure S1: Top-matched WebVision1k instances are chosen: (a) without text enhancement, (b) with text enhancement in VSGraph [3], and (c) with our text enhancement.

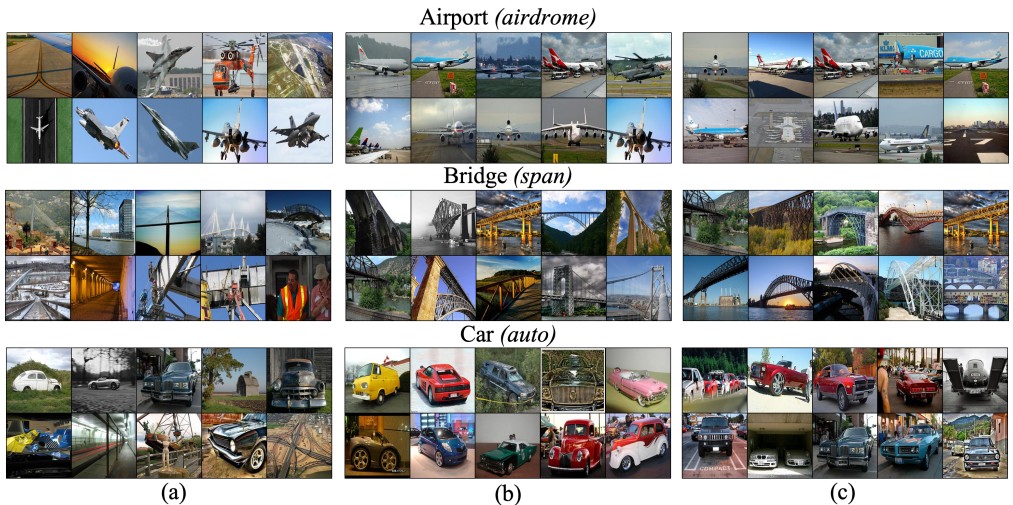

Figure S2: Top-matched NUS-WIDE (Web) instances are chosen: (a) without text enhancement, (b) with text enhancement in VSGraph [3], and (c) with our text enhancement.

---
**Algorithm S1:** CAPro's training procedure.
---
**Data:** Web images and their associated texts and labels $D = \{(\mathbf{x}_i, \mathbf{t}_i, y_i)\}_{i=1}^N$.

1  **Step1 pre-training**
2  **for** $(\mathbf{x}_i, y_i) \in D$ **do**
3     | $\mathcal{L}_i = \mathcal{L}_i^{\text{cls}} + \mathcal{L}_i^{\text{prj}}$;
4     | Update image encoder, classifier, projector, and reconstructor to minimize $\mathcal{L}_i$;
5  **end**
6  **Step2 training**
7  **for** $(\mathbf{x}_i, \mathbf{t}_i, y_i) \in D$ **do**
8     | Extract $\mathbf{v}_i$ from $\mathbf{x}_i$ via the image encoder;
9     | Extract $\mathbf{s}_i$ from $\mathbf{t}_i$ via the text encoder;
10  **end**
11  Build $k$-reciprocal-NN graph $\mathcal{G} = \{\mathcal{V}, \mathcal{E}\}$ with $\{\mathbf{v}_i\}_{i=1}^N$;
12  Enhance text embeddings from $\mathbf{s}_i$ to $\hat{\mathbf{s}}_i$ via graph convolution on $\mathcal{G}$;
13  **for** $c \in \{1, 2, ..., C\}$ **do**
14     | Extract $\mathbf{s}^c$ from $\mathbf{t}^c$ via the text encoder;
15     | **for** $i \in \{1, 2, ..., N | y_i = c\}$ **do**
16        | Match textual instances $\mathbf{s}_i$ to prototypes $\mathbf{s}^c$ to obtain visual anchors $D_K^c$;
17     | **end**
18  **end**
19  Initialize visual prototypes with $D_K$;
20  **for** $(\mathbf{x}_i, y_i) \in D$ **do**
21     | $\mathcal{L}_i = (1 - \lambda^{\text{bts}})\mathcal{L}_i^{\text{cls}} + \lambda^{\text{bts}}\mathcal{L}_i^{\text{bts}} + \lambda^{\text{prj}}\mathcal{L}_i^{\text{prj}} + \lambda^{\text{pro}}\mathcal{L}_i^{\text{pro}} + \lambda^{\text{ins}}\mathcal{L}_i^{\text{ins}}$;
22     | Update image encoder, classifier, and projector to minimize $\mathcal{L}_i$;
23     | Refine $y_i$ to $\hat{y}_i$ to remove noise;
24  **end**
25  **Step3 fine-tuning**
26  **for** $(\mathbf{x}_i, \hat{y}_i) \in D$ **do**
27     | $\mathcal{L}_i = \mathcal{L}_i^{\text{cls}}$;
28     | Update classifier to minimize $\mathcal{L}_i$;
29  **end**
---

Table S3: Effect of $\gamma$ on CAPro without collective bootstrapping.

| $\gamma$ | Reference Provider | Google500 Top1 | Google500 Top5 | ImageNet500 Top1 | ImageNet500 Top5 | NUS-WIDE C-F1 | NUS-WIDE O-F1 | NUS-WIDE mAP |
|---|---|---|---|---|---|---|---|---|
| 0.6 | × | 72.0 | 88.0 | 66.9 | 85.4 | 8.3 | 9.1 | 6.9 |
| 0.8 | × | 71.2 | 87.7 | 65.9 | 84.8 | – | – | – |
| 0.9 | × | – | – | – | – | 39.2 | 44.4 | 46.8 |

## S4.2 Effect of $\gamma$ on Noise Removal