# OpenReview forum: "CAPro: Webly Supervised Learning with Cross-modality Aligned Prototypes"
_NeurIPS.cc/2023/Conference — NeurIPS 2023 poster_

### Official Review · Reviewer_QoQL · 2023-06-21

**Soundness:** 3 good
**Presentation:** 3 good
**Contribution:** 3 good
**Rating:** 5
**Confidence:** 5

**Summary:**

This paper proposes a unified prototypical contrastive learning framework, named Cross-modality Aligned Prototypes (CAPro), to learn visual representations with correct semantics. CAPro exploits web data across modalities to formulate semantically-correct textual and visual prototypes. The authors propose text matching to leverage textual prototypes to establish their noise-robust estimation. They also bring in text enhancement with visual guidance from image neighbors to mitigate the effect of noisy texts on clean sample selection. Further, the authors propose collective bootstrapping (CB) to provide smoother label reference by extending bootstrapping with collective knowledge. Experimental results on WebVision1k and NUS-WIDE (Web) are provided to validate the effectiveness of the proposed method.

**Strengths:**

- The idea of bridging visual and texture prototypes to cope with webly supervised learning is interesting and promising.

- The motivation is clear, and the paper is written well.

- The code is released. The ablation study seems extensive.

**Weaknesses:**

- Eq.(5), (6), and (7) seem very similar to those in [28]. It is expected to discuss the difference between the proposed one with [28]. Why is the proposed one better?

- The total objective loss function shown in Ln 242 contains 4 loss weight hyper-parameters. However, only \lambda_{bts} is discussed in the ablation study. What about other hyper-parameters? Moreover, it is also a concern that the proposed method involves too many hyper-parameters, including these four loss weights, \gamma, update frequency, etc. How to decide the proper value of these hyper-parameters? What about the robustness?

- In Table 1, the result of NCR[58] (i.e., 73.9) seems wrong. The result shown in the NCR paper is 75.7 for NCR and 76.8 for NCR+Mixup+DA, both surpassing the reported result of the proposed method. This raises a concern about the performance of the proposed method.

- Some minor issues:
	- In Ln 62, a period is missing between "selection" and "We".
	- In Ln 69, the sentence uses the past tense, while the former and latter sentences basically use the present tense. It is recommended to change this sentence to present tense to make them consistent.


**Questions:**

N/A

**Limitations:**

The authors have briefly discussed the limitations and broader impact.

---

> ### Author Rebuttal · Authors · 2023-08-08
>
> A4.1 Thank you.
>
> A4.2 First, the Eqs. (5)(6) are exactly the same as Eqs. (1)(4) in MoPro [28] because we adopt the same prototypical and instance-wise contrastive learning.
>
> Second, the control flow in Eq. (7) is inspired by Eq. (5) in MoPro [28], but we differ in that the labels of the top-$K$ matched examples are kept unchanged. Our design guarantees that these top-$K$ samples would provide consistent guidance on noise removal and prototype update, which avoids overfitting in highly noisy classes. Our superior results under both single-label and multi-label scenarios demonstrate that MoPro [28] is prone to overwhelming noise in certain categories, where clean samples would be easily ruled out by the majority of noisy ones. A detailed comparison between our noisy removal policy Eq. (7) and MoPro [28] is in the supplementary (line 119).
>
> Third, our CAPro differs from MoPro in three aspects:
>
> CAPro notices the semantic noise problem and takes advantage of both textual and visual prototypes to handle semantic misalignment. Our visual prototypes are maintained and polished only by semantically-correct examples.
>
> CAPro "creatively" reuses the dictionary (originally set for instance-wise contrastive learning) for collective bootstrapping, where visually-similar neighbors provide label reference by performing dictionary look-up.
>
> CApro, to our best knowledge, is the first to extend prototypical contrastive learning for multi-label classification, where the overwhelming noise ratio and intra-class positive-negative imbalance pose great challenges to optimization. To solve that, CAPro performs prototypical learning in subspaces of the shared embedding space to stabilize training.
>
> A4.3 Please refer to our response A2.5 for the instructions on how to tune the hyper-params. of loss weights in line 242. We will add more explanations on hyper-params. of the loss function.
>
> A4.4 First, ablation studies on $\lambda^{bts}$ and top-$K$ can be found in lines 301-304 of the manuscript. The discussion on threshold $\gamma$ and update frequency is in our supplementary (lines 88, 103).
>
> For $\gamma$ on single-label datasets, its value is related to the percentage of noise in datasets. For WebV1k/Ggl500 ($34\\%$ noise [3]), $\gamma=0.6$ works better than $\gamma=0.8$. For one's own web dataset, if the noise ratio is larger, $\gamma$ should be tuned lower so that wrong labels could be corrected at an earlier stage before overfitting.
>
> For $\gamma$ on multi-label datasets, its value is related to both the percentage of noise and the number ratio of positive-negative samples. For NUS-WIDE ($50\\%$ noise [78] and $0.02$ for avg. ratio of positive-negative examples), $\gamma=0.9$ works better than $\gamma=0.6$. For one's own web dataset, if the noise ratio is smaller and the positive/negative ratio is smaller, $\gamma$ should be tuned higher so that hard positive samples will not be easily discarded to avoid underfitting.
>
> For the update frequency, its value is related to the dataset scale and noise level. For WebV1k/Ggl500, visual prototypes should be updated per epoch to improve their diversity, which better handles the domain gap between web and realistic datasets. For NUS-WIDE, the update frequency could be reduced to stabilize training, where the prototypes can be prone to overwhelming negative examples in each category.
>
> For $\lambda^{bts}$, we suggest $0.1$ would achieve a balance between the individual and collective label references. A much larger value may cause over-smoothing and over-regularization of visual learning.
>
> For top-$K$, its value is related to the percentage of noise. If the noise ratio is less than $30\\%$, $K$ should be set higher than $50$ to include more diverse examples.
>
> The settings of other hyper-params. can be found in the supplementary (Tab. S2). We find current settings work well. For one's own dataset, we believe these values can be set as starting points and finetuned accordingly.
>
> Among all hyper-params., our ablation results show that for $\lambda^{bts}$, $\gamma$, and top-$K$, their values do affect performance and should be set following the rules mentioned above (such as the dataset scale, the noise ratio, and the positive-negative ratio).
>
> For other hyper-params. such as the prototype update frequency, we do not observe significant fluctuation. In other words, the model is robust to these hyper-params.
>
> A4.5 First, the result of $73.9\\%$ for NCR [58] is correct for the batch-size of $256$ and can be found in Tab. 6 in their supplementary [58].
>
> Second, as we pointed out in line 257 (manuscript) and in line 24 (our supplementary), we follow most SOTA methods (see Sec.4.2 in MoPro [28]) to use the standard batch-size of $256$ on the WebV1k dataset.
>
> Third, the improvement by large batch size has been studied by NCR [58] as their results for batch-size=$256$ and batch-size=$1,024$ are respectively $73.9\\%$ and $75.7\\%$. Their vanilla baseline with R50 backbone achieves a top-1 Acc of $74.9\\%$ [58].
>
> Fourth, NCR [58] does not report their top-5 results on WebV1k and does not provide both top-1 and top-5 results on ImgN1k, where we cannot tell if their method can properly handle the domain gap between web and realistic datasets.
>
> Finally, we agree that comparability between different methods should be carefully studied. To help interpret Tab. 1, please see our response to the Reviewer HQEo on how to compare different methods.
>
> We will add results of NCR $\dagger$ [58] ($75.7\\%$ for batch-size of $1,024$) to Tab. 1. More explanations on how to interpret and compare methods will also be added.
>
> A4.6 We correct these sentences accordingly.
>
> A4.7 More discussions on limitations will be added. Please see response A1.9 for details.

---

### Official Review · Reviewer_SPGi · 2023-07-03

**Soundness:** 3 good
**Presentation:** 3 good
**Contribution:** 3 good
**Rating:** 5
**Confidence:** 4

**Summary:**

This paper dives into the study of webly-supervised learning and aims to utilize the neglected alt-text of web images to enhance the learning process. To this end, the authors propose the approach called Cross-modality Aligned Prototypes (CAPro). CAPro is adopted with two modules, namely, text matching&enhacement, and collective bootstrapping. The former aims to assign text to the corresponding prototypes by resorting to the LLMs and cross-modal nearest neighbor mechanism. The latter provides smoother label reference by extending bootstrapping with collective knowledge. Extensive experiments have been conducted on several benchmarks to verify the effectiveness of the proposed CAPro.

**Strengths:**

Most existing webly-supervised learning works mainly focus on the web images and corresponding (noisy) labels while omitting the potential alt-texts (captions). This paper provides a new perspective that uses the alt-text to complement the webly-supervised learning. From this point, the motivation and idea of this paper are novel and interesting.


**Weaknesses:**

1. My major concern is the differences between this work and the existing problem or techniques including the noisy correspondence, noise-robust learning from NNs, and noise removal. First, this paper claims that the existing webly-supervised learning works mainly address certain types of noise including label-flipping noise and out-of-distribution (OOD), while neglecting the semantic noise, namely, the misalignment between image contents and the associated texts. The claim might be correct for the webly-supervised learning community. However, the so-called semantic noise is very similar to the definition of noisy correspondence [61,62, 65]. The authors should provide more discussion to clarify the differences between the so-called noisy correspondence and semantic noise. If the two problems are somewhat similar, I think it would be better to give more discussion on the related works like 'Noisy Correspondence Learning with Meta Similarity Correction'. Second, the differences between the used KNN-graph mechanism and the works in 'Noise-Robust Learning from Neighbors' are encouraged to be further discussed. Third, the proposed noise removal strategy seems to be similar to the sophisticated WSL method (MoPro), which is an important baseline for WSL learning.
2. The performance improvement is limited (See Table 1). This paper adopts a relatively complex pipeline and additional resort to the existing LLMs. However, the performance is marginal compared to the sophisticated WSL baseline (MoPro, ICLR 2020).
2. There are some typos and unclear statements. For example:
i) the definition of 'concept definition texts (Line 56-57)' is lacking
ii) 'with visual guidance from image neighbors (Line 61)' is unclear.
iii) 'selection We' (Line 62)

**Questions:**

My major concern is the  differences between this work and the existing problem and techniques as eloborated on in Weaknesses. Thanks

**Limitations:**

The authors have discussed the limitations.

---

> ### Author Rebuttal · Authors · 2023-08-08
>
> A3.1 Thank you.
>
> A3.2 We would like to explain the differences between our work and previous studies in noisy correspondence learning.
>
> First, the reasons behind these two problems are different.
> The semantic noise is caused by the polysemy retrieval keywords which are used to crawl web images. For example, when we try to retrieve web images of "drumstick" for the percussion music instrument, we may end up with a bunch of images of "drumstick dishes (chicken)" and "drumstick trees (moringa oleifera)". The associated texts of these irrelevant images indeed contain "drumstick", but the image contents are irrelevant to the expected concept.
>
> Noisy correspondence [61, 62, 65] emphasizes the mismatch between an image and its associated text itself. It is mainly caused by mistakes in partitioning the interleaved image and text web data. For example, the caption "a bunch of cows grazing in a dry field together" is wrongly assigned to an image of "giraffes" [65]. Mismatched image-text pairs mainly hinder the performance of cross-modality retrieval.
>
> Second, the solutions to these two problems are different.
>
> The semantic noise belongs to the label noise, which means that web images are incorrectly annotated.  We focus on learning visual representation with categorical noise, which falls under the scenarios of unimodal single-label or multi-label classification.
>
> Noisy correspondence tackles the instance-level mismatched image-text pairs. Its task is to identify alignment errors in the paired data and remove false positives of matching. Most noisy correspondence methods [61, 62, 65] learn to align image and text embeddings to facilitate cross-modal retrieval.
>
> Finally, we agree that there exists an intersection between the two problems. In our webly-supervised learning, we resort to both images and their texts to form semantically-correct visual prototypes. These images and texts can be mismatched and therefore we adopt text enhancement by $k$-reciprocal-NN-based smoothing and reranking to alleviate the noisy correspondence problem. From this point of view, it is indispensable to take noisy correspondence into consideration.
>
> We add another section to the related work:
>
> Noisy Correspondence Rectification
>
> One paradigm similar to WSL might by noisy correspondence rectification or calibration [60,65,62,66,61,63,68,79]. It tackles the mismatched image and text pairs and aims to simultaneously learn aligned visual and textual embeddings for improved cross-modal retrieval. Huang et al. [65] utilizes the memorization effect of neural networks to partition clean and noisy data and then learns to rectify correspondence. Hu et al. [61] derives a unified framework with contrastive learning to reform cross-modal retrieval as an N-way retrieval. Han et al. [66] proposes a meta-similarity correction network to view the binary classification of correct/noisy correspondence as the meta-process, which facilitates data purification.
>
> Although the noisy correspondence removal is closely related to our task, it differs in two aspects: 1) We focus on the label noise where web images are wrongly-annotated by weak keywords or hashtags. Noisy correspondence emphasizes the instance-level mismatch between an image and its associated text. 2) We aim to learn visual representations with correct categorical labels while most methods on noisy correspondence try to align image and text embeddings to improve cross-modal retrieval.
>
> A3.3 More discussions on the differences between previous nearest-neighbor methods and our CAPro will be added to the related work as follows.
>
> It is noted that nearest neighbors play a vital role throughout the components of our CAPro, from text enhancement to text matching and collective bootstrapping. Compared with previous methods of learning from neighbors, our mechanism differs in that:
> 1) We acquire guidance from cross-modality neighbors, where noisy texts are enhanced by image neighbors to alleviate the mismatch problem. In contrast, most previous studies investigate neighbors within one modality.
>
> 2) We exploit reciprocal structures to filter and rerank nearest neighbors for pertinent text matching, while most existing works neglect those top-ranked false positive neighbors.
>
> 3) We resort to neighbors for on-line collective bootstrapping in a manner of dictionary look-up instead of explicit global graph construction.
>
> A3.4 First, CAPro shares the same prototypical contrastive learning with MoPro [28]. The differences between MoPro and our CAPro include:
>
> CAPro notices the semantic noise problem and takes advantage of both textual and visual prototypes to handle semantic misalignment. Our visual prototypes are polished only by semantically-correct examples.
>
> CAPro "creatively" reuses the dictionary for collective bootstrapping, where visually-similar neighbors provide label references by performing dictionary look-up.
>
> CApro, to our best knowledge, is the first to extend prototypical contrastive learning for multi-label classification, where the overwhelming noise and intra-class imbalance pose great challenges. CAPro performs prototypical learning in subspaces of the shared embedding space to stabilize training.
>
> Second, please see the response A4.2 for the differences between our noise removal strategy (Eq. 7) and MoPro (Eq. 5).
>
> Third, the advantage of CAPro over MoPro is highlighted in Fig. 1(c), where performance on $235$ polysemy categories is validated. CAPro indeed improves MoPro by addressing the semantic noise that prevails in these classes. In this case, CAPro can be seen as an effective "booster" to MoPro, which further enables the handling of semantic misalignment with wiser collective bootstrapping. The performance gains are expected to be much more significant in practice usage.
>
> A3.5 We have corrected these typos and statements.
>
> A3.6 Thank you for the insightful comments.
>
> A3.7 Thank you.

---

> > ### Comment · Reviewer_SPGi · 2023-08-18
> > **Replying to the authors' rebuttal**
> >
> > Thanks for the detailed rebuttal. In the rebuttal, the author have clarified the difference between some related works. I would like to maintain my positive rating.

---

### Official Review · Reviewer_XBs5 · 2023-07-04

**Soundness:** 3 good
**Presentation:** 2 fair
**Contribution:** 3 good
**Rating:** 6
**Confidence:** 4

**Summary:**

To handle the label noise problems especially the semantic noise in webly supervised learning, the authors propose a unified prototypical contrastive learning framework named as Cross-modality Aligned Prototypes (CAPro). It exploits web data across modalities to formulate semantically-correct textual and visual prototypes. Besides, collective bootstrapping is proposed to encourage smoother and wiser label reference from appearance-similar instances in a manner of dictionary look-up. Extensive experiments on WebVision1K and NUS-WIDE(Web) demonstrate that the proposed CAPro could handle realistic noise under different scenarios. It also achieves new state-of-the-art performance on open-set recognition.

**Strengths:**

1.	The authors propose a new cross-modality prototypical learning framework named as CAPro, which aims to handle various noises, especially semantic noise.
2.	The proposed CAPro framework is carefully designed and fully explores the cross-modality intervention to filter out noise.
3.	Extensive experiments have been conducted to prove the effectiveness of the proposed method. On the other hand, the code is available and enables reproducing this work.


**Weaknesses:**

1.	The framework figure, i.e., fig. 2, is too complex to be understood. The arrows are not in the same directions, and the symbols are not introduced in caption or texts. There is also lack of explicit module division in the figure. These aspects increase the difficulty to understanding the whole framework.
2.	In experiments, it seems that different methods obtain different “vanilla” performance even with the same backbone, such as R50. Why is the performance different among different methods? Does it mean unfair comparison?
3.	The whole framework is complex since it introduces many modules to handle noise. The whole training process should be organized as explicit algorithm to facilitate the understanding.
4.	There are a lot of hyper-parameters in the total objective. How to choose them during experiments?
5.	There are some typos in the manuscript.


**Questions:**

Please try to address the weaknesses above.

---

> ### Author Rebuttal · Authors · 2023-08-08
>
> More explanations will be added to the manuscript.
>
> A2.1 Thank you for the comments.
>
> A2.2 First, we would like to further explain the function of each module in Fig. 2:
>
> Siamese image encoders: extract features $\mathbf{v}_i$, $\mathbf{v}_i'$ from inputs $\mathbf{x}_i$ and their augmented counterparts $\mathbf{x}_i'$.
>
> Text encoder: generates embeddings $\mathbf{s}_i$, $\mathbf{s}^c$ respectively from the instance $\mathbf{t}_i$ and the category $\mathbf{t}^c$.
>
> Classifier: maps $\mathbf{v}_i$ to predictions $\mathbf{p}_i$ over $C$ classes.
>
> Projector: distills discriminative low-dim.al embeddings $\mathbf{z}_i$ from $\mathbf{v}_i$, followed by $\ell_2$-normalization for unit-sphere constraint on $\mathbf{z}_i$.
>
> Reconstructor: recovers $\tilde{\mathbf{v}}_i$ from $\mathbf{z}_i$ to be close to $\mathbf{v}_i$.
>
> Auxiliary classifier: outputs predictions $\mathbf{q}_i$ on $\mathbf{z}_i$.
>
> Dictionary: records keys for both contrastive learning and collective bootstrapping. The latest embeddings $\mathbf{z}_i'$ are enqueued while the oldest are dequeued.
>
> Second, we will modify Fig. 2 to make it easier to read and follow (see PDF). The refined captions are as follows:
>
> Overview of CAPro. Images $\mathbf{x}_i$ and texts $\mathbf{t}_i$ are respectively fed into the image and text encoders for features $\mathbf{v}_i$ and $\mathbf{s}_i$. Then, $\mathbf{v}_i$ is projected into the embedding space as $\mathbf{z}_i$, followed by the reconstruction from $\mathbf{z}_i$ to $\mathbf{\tilde{v}}_i$. Visual prototypes $\mathbf{z}^c$ are initialized with anchor instances that are selected by matching enhanced texts $\mathbf{\tilde{s}}_i$ to textual prototypes $\mathbf{s}^c$ for semantic alignment. They are constantly polished up by clean images and engage in contrastive learning to constrain cluster distribution. Collective bootstrapping exploits visual dictionary for regularization on the auxiliary classifier output $\mathbf{q}_i$, where each key embedding is matched to the query for the reference $\mathbf{b}_i$. Web labels $y_i$ are simultaneously refined as $\tilde{y}_i$ for ``denoised'' supervision on the classifier output $\mathbf{p}_i$.
>
> A2.3 First, even under the same R50 backbone, different results of "vanilla" baselines are reported [26,28,58,78]. We believe the training settings, especially the batch-size, are the reasons that certain vanilla baselines surpass most SOTAs. In the present study, we follow MoPro [28] to use the standard settings of ImageNet training. Please refer to our supplementary (line 21).
>
> Second, we take VSGraph [78] and NCR [58] as examples to show how the batch-size would affect performance.
> Both VSGraph [78] and NCR [58] adopt the R50 backbone.
>
> But VSGraph, NCR, and their vanilla baselines are trained with a batch size of $1,024$. And their vanilla methods surpass most SOTAs that are trained with a batch-size of $256$.
>
> The benefits of a larger batch-size (1,024 over 256) on WebV1k are studied in NCR [58]. Its top-1 Accs for batch-sizes of $1,024$ and $256$ are respectively $75.7\\%$ and $73.9\\%$.
>
> We believe batch-size is the key factor that affects comparability.
>
> Due to the limited GPU budget, training with a batch-size of $1,024$ is currently not affordable but we are willing to experiment in further research.
>
> Third, we would like to clarify how to fairly interpret Tab. 1:
>
> The comparison between SOTA methods with ours:
>
> Methods in rows 1-5 are not comparable with the proposed CAPro since their backbones are different.
> Methods in rows 6-8 are not comparable due to their optimized training settings.
> Methods in rows 9-18 are all trained with R50 with a batch size of $256$, which are comparable with ours.
>
> The comparison between SOTA methods with their vanilla baselines:
>
> SCC can be compared with the 4th row.
> VSGraph and CoTeach can be compared with the 6th row.
> MoPro can be compared with the 9th row.
> Our CAPro can be compared with the second last row.
>
> A2.4 Thank you for the suggestion. We prepare an algorithm to clearly explain the entire process (see PDF) and will be added to our paper.
>
> A2.5 First, for the total objective (line 242), we follow MoPro [28] to use $\lambda_{pro}=1$ and $\lambda_{ins}=1$. Out of simplicity, we also use $\lambda_{prj}=1$ as the default.
>
> Second, we would like to explain the effect of $\lambda_{pro}$, $\lambda_{ins}$, and $\lambda_{prj}$ on regularzation.
>
> A larger $\lambda_{pro}$ may pull instances too close to their prototypes, which "shrinks" class clusters in the embedding space.
>
> A larger $\lambda_{ins}$ will enforce stronger visual discriminability between two instances. It may cause two examples from the same category to differ greatly and thereafter downgrades the visual prototype update and class cluster regularization.
>
> A larger $\lambda_{prj}$ improves the reconstruction quality of $\tilde{\mathbf{v}}_i$, which encourages $\mathbf{z}_i$ to retain more information of $\mathbf{v}_i$ in the embedding space.
>
> The projection-reconstruction loss is only involved in the pre-training stage (see line 62 in our supplementary), $\lambda_{prj}$ will not affect the prototypical and instance-wise contrastive learning in the following stage.
>
> Third, for custom web dataset, we suggest that $\lambda_{pro}$, $\lambda_{ins}$, and $\lambda_{prj}$ should be tuned according to the performance results under
>
> 1)$\lambda_{pro}=0$ vs. $\lambda_{pro}=1$;
>
> 2)$\lambda_{ins}=0$ vs. $\lambda_{ins}=1$;
>
> 3)$\lambda_{prj}=0$ vs. $\lambda_{prj}=1$.
>
> According to our experiments on both single-label and multi-label datasets, the default settings of $\lambda_{pro}=1$, $\lambda_{ins}=1$, and $\lambda_{prj}=1$ should work well.
>
> For settings of other hyper-params., please refer to our response A4.4.
>
> A2.6 We will double-check grammar and word spelling.
>
> A2.7 We will modify the manuscript accordingly. Please see the point-by-point response above.

---

> > ### Comment · Reviewer_XBs5 · 2023-08-21
> >
> > I appreciate the authors for addressing most of my concerns and questions. Therefore, I will keep my rating.

---

### Official Review · Reviewer_HQEo · 2023-07-06

**Soundness:** 3 good
**Presentation:** 3 good
**Contribution:** 3 good
**Rating:** 6
**Confidence:** 4

**Summary:**

The authors propose a prototypically-aligned contrastive learning framework for vision and language in order to enable better web-scraping of fine-grained, rare concepts that are easily confused or mapped to other more common concepts when either vision or language is considered in isolation (what they term “semantic noise”). Their goal is to improve webly-supervised learning, which is often plagued by label noise, particularly for fine-grained categories. They call their method CAPro, for “Cross-modality Aligned Prototypes.” Their method is built from a few simple ideas: 1) use text prototypes, well-defined in the literature, to scrape a cleaner set of visual prototypes for fine-grained classes, 2) Use visual features to fill gaps or correct errors within text prototypes for fine-grained concepts, generating “semantically-aligned visual prototypes,” 3) an additional cleaning step to further reduce noise between the visual and textual prototypes that uses cluster regularization, 4) “collective bootstrapping” to further smooth concepts or “label references” via essentially an adapted/aggregated/bootstrapped dictionary lookup over the entire dynamic concept dictionary that reduces the effect of examples divergent from the average when making predictions, particularly helpful for visually similar classes.  Their method is highly similar to MoPro, but with the addition of the above described semantic noise correction and regularization across modalities.

After reading the authors rebuttal and discussing with them, as well as reading the comments and discussion with other reviewers, I will increase my score to a 6.

**Strengths:**

The authors show that their method performs well for single-label and multi-labe, on WebVision1k and Imagenet1k, and shows some open-set generalizability. In particular, they define 235 categories from these datasets as exhibiting “polysemy concepts”, and show that their method is particularly effective for these concepts. The gains shown are nice, and the more detailed performance breakdown of the 235 class versions of both datasets is interesting (though it would be nice to also show the performance breakdown for the non-polysemy concepts to help build reader intuition of the impact of polysemy). The method does seem to clearly have an advantage in multi-label challenges, and performs better than CoTeach or VSGraph on open-set concepts. The qualitative examples in table 3 are quite compelling, though they seem potentially highly cherry-picked.

**Weaknesses:**

Notably, their system is quite complex, and feels a bit like a bag of tricks (albeit an effective one), and the method both performs quite similarly to MoPro and often underperforms VSGraph on Top1 (the authors point out that these may not be directly comparable, as they use different backbones, but this makes it difficult to understand or interpret the table. Perhaps they could somehow point out which methods are comparable along which axes in the table? As it is, it’s not very easy to determine the methods value vs other methods). I would also have liked to see more thorough ablations to better understand how impactful their suggested components were vs the additional computational complexity added, to better understand how “worth it” each component would be to use or implement. I find table 4 to be a bit difficult to interpret. I would also like a more thorough analysis of why top-1 suffers, but top-5 benefits (this may be related to over-regularization somehow?).


Nits:
In the abstract: “exacerbates fine-grained visual concept learning.” should perhaps be “exacerbates the challenge of fine-grained visual concept learning.”

Line 26: “The large scale” -> “Large scale”

Line 305: “study” -> “studies”

**Questions:**

I’m curious how this method would handle visual dimorphism in fine-grained classes, for instance dark-eyed juncos which have multiple distinct color morphs, some more common than others. Since the method explicitly seeks prototypical examples, is this at the expense of recognition of anomalies or rarities within-concept? Have the authors seen any interesting failure modes for their method, where simpler methods succeed?

**Limitations:**

The authors have not addressed this at all, which I see as a significant weakness. What are the potential benefits or harms when building prototypical definitions of concepts? What might this mean for instances which do not fit the prototypical visual appearance of a given concept on the internet? How might biases in internet data limit our prototypes?

---

> ### Author Rebuttal · Authors · 2023-08-08
>
> A1.1 We improve Fig. 1 (c) with non-polysemy performance (see PDF).
>
> A1.2 CAPro is indeed a complicated, systematic solution. It is specifically designed to address web noise instead of piling up tricks.
> It follows a similar paradigm of prototypical learning as MoPro [28] and PCL [27].
> Different to them, we take semantic noise into serious consideration.
> Our advantage against MoPro is highlighted in 235 polysemy concepts to demonstrate the benefits of visual-semantic alignment.
>
> A1.3 In Tab. 1, CAPro does not excel VSGraph in top1 but surpasses it in top5 on WebV/ImgN1k. On Ggl/ImgN500, CAPro outperforms it with 7.9%/8.9% in top1. CAPro consistently outperforms VSGraph in Tabs. 2-4 for different benchmarks, tasks, and encoders.
>
> A1.4 First, VSGraph adopts the same R50 but is trained with a batch-size of 1,024. The benefits of a larger batch size are studied in NCR [58]. Its top1 Accs. for 1024 & 256 batch-sizes are 75.7% & 73.9%. We believe batch size is the reason that a baseline [78] surpasses most SOTAs. Due to the limited budget, training with a batch-size of 1024 is currently not affordable, but we will experiment in the future.
>
> Second, we would like to clarify that:
> Methods in the same parts of Tab. 1 are comparable to each other. Methods in rows 1-5 are not comparable to CAPro due to different backbones. Methods in rows 6-8 are not comparable to CAPro due to optimized training settings. CAPro is fairly compared with all methods in rows 9-18.
>
> A1.5 First, we present the params. & GFLOPs of encoders.
> |Encoders|Params.|GFLOPs|
> |---|---|---|
> |R50|25M|3.8|
> |MiniLM|22M|4.7|
> |XLNet|110M|29|
> |GPT-Neo|1.3B|3400|
>
> Second, we present the costs vs. gains for text enhancement, where N is the # of nodes, k is # of neighbors per node, and $d_v$ is the dim. of $\mathbf{v}$.
> |Text Encod.|Text Enhance.|Cost|Ggl500 Top1|ImgN500 Top1|
> |---|---|---|---|---|
> |MiniLM/XLNet/GPT-Neo|VSGraph|$O(N^2d_v)$+$O(N{d_v}^2+Nkd_v)$|72.0/71.6/72.0|66.9/66.8/67.2|
> |MiniLM/XLNet/GPT-Neo|Ours|+$O(3Nkd_v)$+$O(4k)$+$O(4klog(4k))$|+3.5/+3.8/+3.7|+4.6/+4.7/+4.4|
>
> With k=5 & k=10 for WebV1k & NUS-WIDE, our improvement over VSGraph is worthy at the expense of such a low cost.
>
> Third, we present the costs vs. gains for reference provider, where m is the batch size, d_v is the dim. of v, Q is the size of dictionary, d_p is the dim. of z, and C is the # of classes.
> |Text Encod.|Ref. Provider|Cost|Ggl500 Top1|ImgN500 Top1|
> |---|---|---|---|---|
> |MiniLM|MixUp/BootStrap/LabelSmooth/SCC|--|75.7/75.5/75.4/73.8|71.4/71.3/71.2/70.2|
> |MiniLM|NCR| $O(m^2(d_v+C))$|-0.2/+0/+0.1/+1.7|+0.1/+0.2/+0.3/+1.3|
> |MiniLM|Our CB| $O(mQ(d_p+C))$|+0.3/+0.5/+0.6/+2.2|+0.6/+0.7/+0.8/+1.8|
>
> Operations of our CB are fast to compute since PyTorch supports efficient matrix multiplication on GPUs.
> Our $d_p$=128 is 16x << $d_v$=2048 in NCR & our m=256 is 4x << m=1024 in NCR.
> For WebV1k, our cost is 1.35x < NCR.
> For NUS-WIDE, our cost is 20.37x << NCR.
> It is reasonable to conclude that our CB is more efficient & effective than NCR.
>
> Finally, text encod.& enhance. are off-line and executed only once. They do not participate in network optimization. Besides, the pretrained text encoders are only used for inference under 1 V100 GPU. Therefore, the additional cost is acceptable in return for semantically-correct web images.
>
> A1.6 First, for Tab. 4, methods in rows 2-7 outperform the 1st row with better top1&top5, validating the textual knowledge. For different text encoders, we outperform VSGraph in both top1 & top5.
>
> Second, SCC estimates confidence independently and neglects the relationship between an instance and its prototype. NCR brings no gains. Its effect is limited without a large batch size. Bootstrap & label smooth degrades both top1 & top5 on ImgN500 while mix-up benefits top1 on Ggl500.
>
> Mix-up improves CAPro in top1 but lowers top5. It adopts convex combinations for both inputs & targets, enforcing a stronger regularization than our CB where we only recombine targets.
> For WebV1k, examples with noisy labels still resemble their prototypes and therefore neighbor knowledge brings useful reference. Mix-up does not consider appearance similarity and causes over-regularization.
>
> A1.7 Thank you. We fix typos.
>
> A1.8 First, CAPro can handle fine-grained categories on WebV1k. The introduction of atypicals increases the risk of noise. For generalization on anomalies or rarities, one solution is to choose both top-K and randomly sampled instances.
>
> Second, for WebV1k, both MoPro&CAPro underperform the vanilla on a total of 387&312 classes. Top5 failures: screen, sunGlasses, bellCote, ballPlayer, popBottle. For ImgN1k, MoPro&CAPro underperform the vanilla on a total of 450&358 classes. Top-5 failures: silkyTerrier, walkerHound, academicGown, standardSchnauzer, bellCote.
>
> Findings:
>
> 1) Domain gap exists between web and realistic datasets.
>
> 2) The vanilla tends to overfit training set so that it outperforms on highly similar concepts: screen & monitor, sunGlasses & sunGlass.
>
> Mistakes on silky & yorkshire Terrier, walker & englishFox hound are ascribed to over-regularization. The inter-class relationship might be used for class-wise adjustment.
>
> A1.9 Two benefits:
>
> Noisy web data can be corrected by measuring the instance-prototype distance.
> Inter-class relationship can be statistically studied to shed light on similarities between species.
>
> Three drawbacks:
>
> Tolerance to the intra-class minority. Web data follow long-tailed distribution. The more common one instance is, the greater the likelihood that it gets exposed.
> =>to introduce randomness into init. & update of prototypes.
>
> The domain bias of web data. Their styles (advert. & render.) are different from realistic ones. Specific modalities (infrared & CT) are unavailable.
> =>to prepare realistic images for guided-training.
>
> Prior knowledge about class hierarchy. Coarse-grained or improper descriptions about hierarchical structure would devalue semantic alignment.
> =>to perform a thorough analysis of concepts.

---

> > ### Comment · Reviewer_HQEo · 2023-08-19
> > **Thank you for the clarification and analysis in your rebuttal**
> >
> > I thank the authors for their rebuttal, and I believe that with the additions they have proposed in the general rebuttal across all reviews the paper will be significantly clarified and strengthened which is a win for the peer review process and will lead to a notably improved manuscript. If all 1-10 of the promised improvements are incorporated thoroughly and well into the final manuscript I believe that would justify an increased score of 6.

---

### Author Rebuttal · Authors · 2023-08-08

Dear reviewers, area chairs, and senior area chairs,

We sincerely thank that all four reviewers are positive towards our paper, and provide detailed, constructive comments.

According to these comments and suggestions, we will modify our manuscript by:

1) improving all figures to make them easy to understand and follow;

2) adding more explanations on the differences between our CAPro with MoPro;

3) adding another section into the related work (Noisy Correspondence) to introduce the differences between webly supervised learning and noisy correspondence removal;

4) adding more explanations on the differences between our work and previous methods of learning from noisy labels;

5) adding computational complexity to the ablation study to show the additionally introduced cost;

6) adding more explanations on how to interpret the comparability between SOTA methods when different backbones and training settings are adopted;

7) adding more discussions on how to tune hyper-parameters;

8) adding more discussions on our findings of failure cases;

9) adding more discussions on the limitations;

10) polishing English writings.

Please see the point-by-point response below for each reviewer.

Besides, we provide additional **tables, figures, and algorithms** in the uploaded **one-page PDF** for better explanations. These tables, figures, and algorithms will be magnified in the manuscript, but have to be compressed now to fit into one page only for rebuttal.

Finally, we would like to express our gratitude again for all the valuable comments that would significantly help improve the quality of the manuscript.

---

> ### Comment · Area_Chair_8VGL · 2023-08-18
>
> thanks, all responses have been read and will be taken into account

---

### Decision · Program_Chairs · 2023-09-21

**Decision:**

Accept (poster)

**Comment:**

There are only positive scores, two weak accept and two borderline accept (considering decisions to increase BA->WA in the comments not reflected in the review form). The reviewers praise the motivation, method design and effectiveness, and the scope of the experiments. Concerns are about complexity and results, but three out of four reviewers explicitly comment that they read the rebuttal and maintain their positive rating. Answers to the fourth reviewer (who did not respond to the rebuttal) seem reasonable.